

# Adjoint inversion of Chinese non-methane volatile organic compound emissions using space-based observations of formaldehyde and glyoxal

Hansen Cao[1], Tzung-May Fu[1, *], Lin Zhang[1], Daven K. Henze[2], Christopher Chan Miller[3], Christophe Lerot[4], Gonzalo González Abad[3], Isabelle De Smedt[4], Qiang Zhang[5], Michel van Roozendael[4], Kelly Chance[3], Jie Li[6], Junyu Zheng[7], Yuanhong Zhao[1]

[1]Department of Atmospheric and Oceanic Sciences and Laboratory for Climate and Ocean-Atmosphere Studies, School of Physics, Peking University, Beijing,100871, China
[2]Department of Mechanical Engineering, University of Colorado, Boulder, USA
[3]Atomic and Molecular Physics Division, Harvard-Smithsonian Center for Astrophysics, Cambridge, Massachusetts, USA
[4]Belgian Institute for Space Aeronomy (BIRA-IASB), Brussels, Belgium
[5]Center for Earth System Science, Tsinghua University, Beijing, China
[6]Institute of Atmospheric Physics, Chinese Academy of Sciences, Beijing, China
[7]College of Environmental Science and Engineering, South China University of Technology, Guangzhou, China

*Correspondence to*: Tzung-May Fu (tmfu@pku.edu.cn)

**Abstract.** We used the GEOS-Chem model and its adjoint to quantify Chinese non-methane volatile organic compound (NMVOC) emissions for the year 2007, using the vertical column concentrations of formaldehyde and glyoxal observed by the Global Ozone Monitoring Experiment-2A (GOME-2A) instrument and the Ozone Monitoring Instrument (OMI) as constraints. We conducted a series of inversion experiments using different combinations of satellite observations to explore the impacts on top-down emission estimates due to different satellite retrievals. Our top-down estimates for Chinese annual total NMVOC emission was 23.4 to 35.4 (average 30.8) Tg C y$^{-1}$, including 13.5 to 19.7 (average 17.0) Tg C y$^{-1}$ from anthropogenic sources, 8.9 to 14.8 (average 12.6) Tg C y$^{-1}$ from biogenic sources, and 1.1 to 1.5 (average 1.2) Tg C y$^{-1}$ from biomass burning. In comparison, the most widely-used bottom-up estimate for Chinese annual total NMVOC emission was 27.4 Tg C y$^{-1}$, including 15.5 Tg C y$^{-1}$ from anthropogenic sources, 10.8 Tg C y$^{-1}$ from biogenic sources, and 1.1 Tg C



y$^{-1}$ from biomass burning. The simultaneous use of glyoxal and formaldehyde observations helped
distinguish the NMVOC species from different sources and was essential in constraining anthropogenic
emissions. Our four inversions consistently showed that the emissions of Chinese anthropogenic
NMVOC precursors of glyoxal were larger than the *a priori* estimates. Our top-down estimates for the
Chinese annual emission of anthropogenic aromatics (benzene, toluene, and xylene) ranged from 5.0 to
7.3 Tg C y$^{-1}$, 2% to 49% larger than the estimate of the bottom-up inventory (4.9 Tg C y$^{-1}$). Model
simulations using the average of our top-down NMVOC emission estimates showed that surface
afternoon ozone concentrations over northern and central China increased 5-12 ppb in June and
decreased 5-13 ppb in December relative to the simulations using the *a priori* emissions and were in
better agreement with measurements. We concluded that the satellite observations of glyoxal and
formaldehyde together provided quantitative constraints on the emissions and source types of
NMVOCs over China and improved our understanding on regional chemistry.
**1    Introduction**
Non-methane volatile organic compounds (NMVOCs) are emitted into the atmosphere from surface
anthropogenic, biogenic, and biomass burning sources. NMVOCs are precursors to tropospheric ozone
and secondary organic aerosols, both of which are climate forcers and major air pollutants. NMVOC
also affect the oxidation capacity of the atmosphere, which in turn changes the lifetimes of greenhouse
gases and other pollutants (Monks, 2005; Lelieveld et al., 2008). It is thus crucial to quantify NMVOC
emissions in order to understand their impacts on atmospheric chemistry and climate on both global
and regional scales. Here we used satellite observations and a chemical transport model to constrain
NMVOC emissions from China and assessed their impacts on seasonal surface ozone.

Emissions of trace species are traditionally estimated in a "bottom-up" manner using activity data and
emission factors, but these bottom-up estimates are sometimes susceptible to large uncertainties. This
is especially true for NMVOC emissions in developing countries such as China, because (1) a wide
range of species, source activities, and technologies are involved (Q. Zhang et al., 2009; Kurokawa et
al., 2013; Li et al., 2014; Qiu et al., 2014), (2) locally-representative emission factors are often not
measured (Wei et al., 2008; Zhao et al., 2011), and (3) reliable activity data are often incomplete,





particularly for small-scale industries, residential activities, and agricultural waste burning (Q. Zhang
et al., 2009). Table 1 shows bottom-up estimates for Chinese total annual NMVOC emissions for the
years 2005 to 2012, which ranged from 21.6 to 51.7 Tg y$^{-1}$ (Guenther et al., 2006; Bolscher et al., 2007;
Bo et al., 2008; Q. Zhang et al., 2009; van der Werf et al., 2010; Cao et al., 2011; Huang et al., 2012;
Kurokawa et al., 2013; Li et al., 2014; Wu et al., 2016). The large uncertainties in these Chinese
NMVOC emission estimates have led to great difficulty in evaluating their impacts on regional
chemistry (Han et al., 2013; Wang et al., 2014).

A complementary, "top-down" approach of quantifying emissions uses observations of the targeted
species or its chemical derivatives, combined with a chemical transport model acting as a transfer
function, to invert for the fluxes of the targeted species. In particular, tropospheric column
concentrations of formaldehyde (HCHO), retrieved from satellite UV-backscatter observations, have
been used to constrain NMVOC emissions. Formaldehyde is produced at high yields during the
oxidation of many NMVOC species, as well as emitted directly from anthropogenic and biomass
burning activities (Akagi et al., 2011). Early applications of satellite-observed formaldehyde columns
mainly focused on areas where the local NMVOC fluxes were dominated by biogenic emissions during
the growing season, such as the southeast U.S. (Palmer et al., 2003, 2006; Millet et al., 2006, 2008),
Europe (Dufour et al., 2009; Curci et al., 2010), the Amazon (Barkley et al., 2008, 2009, 2013), and the
tropical central Africa (Marais et al., 2012, 2014a). These studies showed that the observed high
concentrations of formaldehyde over densely-vegetated areas were linearly proportional to the local
biogenic isoprene flux during the growing season.

Later studies constrained NMVOC emissions from multiple sources by analyzing the spatiotemporal
variability of the observed formaldehyde columns (Shim et al., 2005; Fu et al., 2007; Stavrakou et al.,
2009b; Curci et al., 2010; Gonzi et al., 2011; Marais et al., 2014b; Zhu et al., 2014). Fu et al. (2007), a
forerunner of this study, analyzed the spatial and seasonal variation of the formaldehyde column
observations from the Global Ozone Monitoring Experiment (GOME) over East and South Asia. They
showed that, during the early 2000s, Chinese reactive NMVOC fluxes from biogenic, anthropogenic,
and biomass burning sources were 3, 1.2, and 8.8 times their respective bottom-up estimates. In
particular, Fu et al. (2007) found a large, annually-recurring NMVOC source over the North China



Plain (NCP) in June, which they attributed to crop residue burning after the local harvest of winter
wheat. However, these top-down studies using only formaldehyde as constraints relied on bottom-up
statistics to differentiate between NMVOC source types.

More recently, satellite measurements of tropospheric glyoxal columns emerged as an additional
constraint on NMVOC emissions (Stavrakou et al., 2009a). Like formaldehyde, glyoxal is produced
during the oxidation of many NMVOCs (including most importantly isoprene), as well as emitted
directly from biomass burning (Fu et al., 2008; Myriokefalitakis et al., 2008). In addition, glyoxal is
produced at high yields at the initial ring-cleaving stage during the oxidation of aromatics (Volkamer,
2001; Nishino et al., 2010), which are mainly anthropogenic. In contrast, formaldehyde production
from the oxidation of aromatics is further downstream and thus spatially diffused (Volkamer, 2001). As
such, simultaneous analyses of formaldehyde and glyoxal observations can help differentiate between
biogenic and anthropogenic NMVOC emissions. Stavrakou et al. (2009a) pioneered a two-compound
inversion using tropospheric glyoxal and formaldehyde column observations from the SCIAMACHY
satellite instrument to constrain the global sources of glyoxal. They estimated that the anthropogenic
NMVOC fluxes over East Asia for the year 2005 were a factor of 2-3 larger than the bottom-up
estimates of the Emission Database for Global Atmospheric Research (EDGAR, v3.3) inventory
(Olivier et al., 2001, 2002) and the REanalysis TROpospheric (RETRO) emission inventory (Schultz et
al., 2007). In addition, they inferred a large missing source of glyoxal over the global continents, which
they attributed to production from an unknown biogenic precursor.

Over eastern China, Liu et al. (2012) showed that the glyoxal column concentrations observed by
SCIAMACHY in August 2007 was more than twice the simulated glyoxal columns using the
bottom-up emission inventory developed by Q. Zhang et al. (2009). Over the Pearl River Delta area
(PRD) in southern China, the discrepancy was at least a factor of three. They suggested that the
missing glyoxal source over eastern China was anthropogenic, on the basis that the anonymous
glyoxal columns observed by SCIAMACHY were spatially correlated with anthropogenic $NO_x$
emissions. Their estimated Chinese anthropogenic aromatics emission was 13.4 Tg $y^{-1}$, which was six
times the 2.4 Tg $y^{-1}$ aromatic flux estimated by Q. Zhang et al. (2009). In contrast, Chan Miller et al.
(2016) simulated the formaldehyde and glyoxal column concentrations over the Pearl River Delta area



(PRD) in southern China for the years 2006 and 2007 using the same inventory developed by Q.
Zhang et al. (2009). They found that their simulated formaldehyde columns were consistent with the
OMI formaldehyde observations, while their simulated glyoxal columns were lower than OMI
observations by only 40%. They attributed the high anthropogenic aromatics emission estimate by Liu
et al. (2012) to a regional high-bias in the SCIAMACHY data, as well as underestimated yields of
glyoxal from the oxidation of aromatics.

One limitation in the use of satellite observations of formaldehyde and glyoxal for constraining
NMVOC sources is their inherent uncertainty. Several studies have compared GOME-2A and OMI
formaldehyde column observations against aircraft or ground-based measurements at a few locations
around the world (De Smedt et al., 2015; Lee et al., 2015; Wang et al., 2017; Zhu et al., 2016). Zhu et
al. (2016) compared the GOME-2A-observed formaldehyde column concentrations over the Southeast
U.S. in summer 2013 against aircraft measurements and found the satellite measurements to be too
low by a factor of approximately 1.7. Chan Miller et al. (2017) found that glyoxal column
concentrations observed by OMI were lower than the aircraft measurements over the Southeast U.S. in
summer 2013 by a factor of 1.5. Wang et al. (2017) compared the bi-monthly mean GOME-2A and
OMI formaldehyde column concentrations retrieved by De Smedt et al. (2012, 2015) against
ground-based multi-axis differential optical absorption spectroscopy (MAX-DOAS) measurements at
a rural site in eastern China. They found that both satellite retrievals were systematically lower than
the ground-based measurements by approximately 20%. These studies highlight the potential impacts
on top-down NMVOC emission estimates due to uncertainty associated with satellite retrievals.

In this study, we used satellite retrievals of both formaldehyde and glyoxal, along with an updated
chemical transport model and its adjoint, to constrain NMVOC emissions from China for the year
2007. We conducted sensitivity experiments to evaluate the impacts on the top-down estimates due to
different satellite retrieval constraints, with the goal of obtaining a most probable range of top-down
estimates. Finally, we examined the impacts of our top-down NMVOC emission estimates on surface
ozone concentrations over China.



## 2   Model and data

### 2.1 The GEOS-Chem model and its adjoint

We used the GEOS-Chem global 3D chemical transport model (version 8.2.1) to simulate the emission, transport, chemistry, and deposition of NMVOCs, as well as the resulting formaldehyde and glyoxal column concentrations for the year 2007. GEOS-Chem was driven by the assimilated meteorological data from the NASA Goddard Earth Observing System (GEOS-5) (Bey et al., 2001). To drive our simulations, the horizontal resolution of GEOS-5 data was downgraded from its native $2/3\,°$ longitude $\times$ $1/2\,°$ latitude to $5\,°$ longitude $\times 4\,°$ latitude. The number of vertical levels was reduced from 72 to 47 by merging layers in the stratosphere. The lower 2 km of the atmosphere was resolved by 14 levels. The temporal resolution of GEOS-5 data into GEOS-Chem is 3 h for atmospheric variables and 1 h for surface variables.

We updated the dicarbonyl chemical mechanism in GEOS-Chem developed by Fu et al. (2008), which in turn was originally adapted from the Master Chemical Mechanism (MCM) version 3.1 (Saunders et al., 2003; Bloss et al., 2005). NMVOC precursors of formaldehyde in our mechanism included ethane, propane, $\geq C_4$ alkanes, ethene, $\geq C_3$ alkenes, toluene, xylenes, isoprene, and monoterpenes. NMVOC precursors of glyoxal in our mechanism included propane, alkanes, ethene, $\geq C_3$ alkenes, ethyne, benzene, toluene, xylenes, isoprene, monoterpenes, glycolaldehyde, and 2-methyl-3-bute-2-nol. OH-oxidation of isoprene is a major source of both formaldehyde and glyoxal over China (Fu et al., 2007, 2008; Myriokefalitakis et al., 2008). We replaced the isoprene photochemical scheme with that used in GEOS-Chem v10.1 (Paulot et al., 2009a,b; Mao et al. 2013), where formaldehyde and glyoxal were produced from isoprene oxidation via the $RO_2+NO$ pathway under high-$NO_x$ conditions and via $RO_2$-isomerization under low-$NO_x$ conditions. Li et al. (2016) compared the productions of formaldehyde and glyoxal from isoprene oxidation in this updated scheme with those in the MCM version 3.3.1 (Jenkin et al., 2015). They showed that the production pathways and yields of formaldehyde and glyoxal were similar in the two schemes under the high-NOx conditions typical of eastern China.



We updated the molar yields of glyoxal from the OH oxidations of benzene (33.3%), toluene (26.2%),
and xylenes (21.0%) following the latest literature (Arey et al., 2009; Nishino et al., 2010). These new
molar yields were higher than those used in Fu et al. (2008) but still lower than those used by Chan
Miller et al. (2016) (75% for benzene, 70% for toluene, 36% for xylenes), which were taken from MCM
version 3.2 (Bloss et al., 2005). In MCM version 3.2, more than half of the glyoxal from aromatics
oxidation were produced during second- and later-generation photochemistry, but such productions are
still uncertain, with limited experimental support (Bloss et al., 2005).

Formaldehyde and glyoxal in our model were both removed by photolysis, as well as dry and wet
deposition (Fu et al., 2008). We updated the Henry's law constant for glyoxal (Ip et al., 2009) and
added the dry deposition of formaldehyde, glyoxal, methyglyoxal and glycolaldehyde on leaves (Mao
et al., 2013). In addition, we assumed that glyoxal was reactively uptaken by wet aerosols and cloud
droplets with an uptake coefficient $\gamma = 2.9 \times 10^{-3}$ (Liggio et al., 2005; Fu et al., 2008). All other physical
and chemical processes in our forward model were as described in Fu et al. (2008).

For the forward model described above, we developed the adjoint by modifying the standard
GEOS-Chem adjoint (version 34) (Henze et al., 2007). We used the Kinetic PreProcessor (KPP)
(Daescu et al., 2003; Sandu et al., 2003) to construct the adjoint of the updated photochemical
mechanism. Adjoint algorithms were updated to include the emission and deposition processes of
formaldehyde and glyoxal precursors. The aqueous uptake rate of glyoxal by wet aerosols was a
function of the ambient glyoxal concentration and the total wet aerosol surface area (Fu et al., 2008).
We linearized this uptake process by archiving the wet aerosol surface areas in the forward simulations
for use in the backward integrations.

We verified the adjoint model mathematically in two ways. Firstly, we used the adjoint to calculate the
sensitivities of global glyoxal and formaldehyde burdens to biogenic isoprene and anthropogenic
xylenes emissions, respectively, and found that the results reproduced the calculated sensitivities from
the forward model (Figure S1 in Supplementary Information). Secondly, we used an *a priori* NMVOC
emission inventory (Section 2.2) to drive the forward model and took the resulting global tropospheric
formaldehyde and glyoxal column concentrations as pseudo observations. We then used the pseudo



observations of formaldehyde and glyoxal, respectively, to successfully optimize back to close to the *a*
*priori* NMVOC emission estimates over high-emission areas from an initial emission guess that was
five times larger (Figure S2 in Supplementary Information). These experiments demonstrated the
usefulness of the adjoint model for the inversion of NMVOCs emissions.

**2.2 *A priori* emission estimates of Chinese NMVOCs**
As a starting point for our inversion, we used the most widely-used NMVOC emission estimates for
China as the *a priori*. Table 2 summarizes the annual total of these *a priori* emission estimates and their
associated uncertainties.

The *a priori* biogenic NMVOC emissions from China and from the rest of the world were calculated
with the MEGAN algorithm (Guenther et al., 2006) and dependent on temperature, shortwave radiation,
and monthly mean leaf area index. The annual total biogenic NMVOC emissions over China for the
year 2007 was 10.8 Tg C $y^{-1}$, including 6.6 Tg C $y^{-1}$ of isoprene. Previous estimates of Chinese
biogenic NMVOC emissions ranged from 5.0 to 11.0Tg C $y^{-1}$ (Guenther et al., 2006; Fu et al., 2007;
Stavrakou et al., 2015). Based on this range, we estimated the uncertainty of the *a priori* biogenic
emissions over China to be ±55%.

The *a priori* emissions for Chinese anthropogenic NMVOCs were from the Multi-resolution Emission
Inventory for China (MEIC) inventory (Li et al., 2014), developed for the year 2010 at 0.25 ° × 0.25 °
resolution. The MEIC inventory, including emissions from industry, transportation, power generation
and residential activities, was compiled using monthly Chinese provincial activity data and a
combination of Chinese and western emission factors. The estimated annual Chinese anthropogenic
NMVOC emission was 15 Tg C $y^{-1}$, including 64% from industries, 24% from residential activities, 10%
from transportation, and 1% from power generation. The estimated annual Chinese anthropogenic
emission of aromatics was 4.9 Tg C $y^{-1}$, including 73% from industries, 15% from residential activities,
9% from transportation, and 3% from power generation. Previous estimates of Chinese anthropogenic
NMVOC emissions for the years 2005 to 2012 ranged from 10.7 to 29.8 Tg C $y^{-1}$, with aromatics
emissions ranging from 2.1 to 11.3 Tg C $y^{-1}$ (Bo et al., 2008; Q. Zhang et al., 2009; Cao et al., 2011;





Liu et al., 2012; Kurokawa et al., 2013; Li et al., 2014; Stavrakou et al., 2015; Wu et al., 2016). We
therefore estimated the uncertainty for the *a priori* Chinese anthropogenic NMVOC emission estimates
to be ±200%. Anthropogenic NMVOC emissions for the rest of the Asia were from the inventory
compiled by Li et al. (2017) for the year 2010. Anthropogenic NMVOC emissions for Europe, U.S.,
and the rest of the world were from the European Monitoring and Evaluation Programme (EMEP)
inventory (Vestreng, 2003), the U.S. EPA 2005 National Emission Inventory (NEI05) (Brioude et al.,
2011; Kim et al., 2011), and the Emission Database for Global Atmospheric Research (EDGAR)
inventory (version 2.0) (Olivier et al., 1999), respectively, and scaled to the year 2007 using $CO_2$
emissions (van Donkelaar et al., 2008).

Post-harvest, in-field burning of crop residue has been recognized as a large seasonal source of
NMVOCs in China (Fu et al., 2007; Huang et al., 2012; Stavrakou et al., 2016), but this emission has
been severely underestimated in inventories based on satellite burnt area observations (Liu et al., 2015).
Huang et al. (2012) estimated the Chinese CO emission from crop residue burning to be 4.0 Tg y$^{-1}$,
based on MODIS daily thermal anomalies, Chinese provincial burnt-biomass data, and emission factors
from Akagi et al. (2011). We scaled this CO flux using speciated NMVOC emission factors from crop
residue burning from the literature (Hays et al., 2002; Akagi et al., 2011) and then multiplied the
resulting NMVOC flux estimate by two. The reason for doubling the scaled NMVOC flux was that the
emission factors for many NMVOC species were not measured, such that the sum of the speciated
NMVOC emission factors was only half of the measured total NMVOC emission factor (Akagi et al.,
2011). This difference may partially explain why the top-down study by Stavrakou et al. (2016) using
satellite observations of formaldehyde found that Huang et al. (2012) underestimated the NMVOC flux
from crop fires over the North China Plain (NCP) in June by at least a factor of two.

Our resulting *a priori* estimate for Chinese annual NMVOC emissions from biomass burning was 1.1
Tg C y$^{-1}$, including 0.86 Tg C y$^{-1}$ from crop residue burning (obtained by scaling Huang et al., 2012)
and 0.24 Tg C y$^{-1}$ from other types of biomass burning activities (taken from the Global Fire Emissions
Database version 3, GFED3) (van der Werf et al., 2010). Previous estimates of Chinese NMVOC
emissions from biomass burning for the years 1996 to 2012 ranged from 0.24 to 3.2 Tg C y$^{-1}$ (Fu et al.,
2007; van der Werf et al., 2010; Wiedinmyer et al., 2011; Huang et al., 2012; Liu et al., 2015;



Stavrakou et al., 2015, 2016). We therefore assigned an uncertainty of ±300% to the *a priori* Chinese
biomass burning NMVOC flux. Biomass burning emissions from the rest of the world were taken from
GFED3 (van der Werf et al., 2010).
Figure 1 (a)-(c) show the spatial distribution of the *a priori* Chinese NMVOC emissions from biomass
burning, anthropogenic, biogenic, and total sources, respectively. Biomass burning emissions were
highest over the NCP and southwest China, reflecting strong emissions from crop residue burning over
the NCP in June and from land-clearing burning over southwest China during February to April,
respectively. Chinese anthropogenic and biogenic NMVOC sources both showed a general west-to-east
gradient, following population and vegetation densities. Biogenic NMVOC emissions reflected the
combined modulation by vegetation densities, temperature, and sunlight. Anthropogenic NMVOC
fluxes exceeded $10^3$ kg C km$^{-2}$ y$^{-1}$ throughout the industrialized and densely populated eastern China,
with the highest fluxes over the NCP and around the Yantze River Delta area.
Figure 2 shows the seasonal variation of the *a priori* Chinese NMVOC emissions. The *a priori*
anthropogenic NMVOC fluxes were higher during the cold months and lower during the warm months,
driven by the seasonal strengths of industrial and residential activities (Li et al., 2017). The *a priori*
biogenic NMVOC fluxes showed the opposite seasonal pattern, with more than half of the total annual
flux emitted in summer. The *a priori* biomass burning NMVOC source was relatively small except
when it peaked in June due to the burst of post-harvest burning over the NCP and in spring due to
land-clearing over southwest China. As a result, the Chinese NMVOC emissions were predominantly
anthropogenic in January but mainly biogenic in June. During the transition months of April and
October, the anthropogenic and biogenic contributions to the total NMVOC emissions were
comparable.

### 2.3 Formaldehyde and glyoxal column concentrations observed by GOME-2A and OMI

We used the monthly mean tropospheric formaldehyde and glyoxal column concentrations retrieved
from the Global Ozone Monitoring Experiment-2A (GOME-2A) instrument and the Ozone
Monitoring Instrument (OMI) for the year 2007 to constrain Chinese NMVOC sources. The technical



details of these four sets of satellite retrievals are summarized in Table 3.

The native GOME-2A pixel vertical column densities (VCDs) of formaldehyde and glyoxal were
retrieved by De Smedt et al. (2012) and Lerot et al. (2010), respectively, using protocols briefly
described below. First, pixel slant column densities (SCDs) of formaldehyde and glyoxal were
retrieved in the 328.5-346 nm and 435-460 nm windows, respectively, using the Differential Optical
Absorption Spectroscopy (DOAS) technique (Platt et al., 1979). Previous glyoxal SCD retrievals often
showed biases over remote tropical oceans due to absorption from liquid water (Vrekoussis et al., 2010;
Wittrock et al., 2006; Lerot et al., 2010). This bias was corrected in Lerot et al. (2010) by explicitly
accounting for liquid water absorption during the DOAS fitting. Second, pixel SCDs were converted
into VCDs using air mass factors (AMF), which was calculated using Linearized Discrete Ordinate
Radiative Transfer model (LIDORT) (Spurr, 2008) and trace gas profiles simulated by the IMAGE v2
model (Stavrakou et al., 2009b). The native pixel VCDs were gridded to daily means at $0.25\degree \times 0.25\degree$
resolution (De Smedt et al., 2012; Lerot et al., 2010). We further averaged the daily means to monthly
means at $5\degree$ longitude $\times 4\degree$ latitude resolution. The retrieval errors of the spatially-and-temporally
averaged VCDs were estimated to be 30%-40% for formaldehyde and 40% for glyoxal, due to a
combination of errors associated with the SCD retrievals, the reference sector correction, the *a priori*
profile, and the AMFs (De Smedt et al., 2012; Lerot et al., 2010).

The OMI native pixel VCDs of formaldehyde and glyoxal were retrieved by González Abad et al.
(2015) and Chan Miller et al. (2014), respectively. Briefly, formaldehyde and glyoxal pixel SCDs were
retrieved by directly fitting the absorption spectra in the 328.5 – 356.5 nm (formaldehyde) and 435 –
461 nm (glyoxal) windows, respectively (Chance, 1998; Kurosu et al., 2004, 2007; Chan Miller et al.,
2014), and then converted to pixel VCDs using AMF calculated with a linearized vector discrete
ordinate radiative transfer model, VLIDORT (Spurr, 2006), and trace gas profiles simulated by the
GEOS-Chem model (González Abad et al., 2015). Liquid water absorption was also explicitly
calculated for the glyoxal retrieval (Chan Miller et al., 2014). The typical uncertainties of
OMI-observed pixel VCDs over polluted areas were estimated to be 30% to 45% for formaldehyde
and 104% for glyoxal (González Abad et al., 2015; Chan Miller et al., 2014). The native pixel VCDs
were averaged to monthly means at $5\degree$ longitude $\times 4\degree$ latitude resolution. For glyoxal, we further



removed VCDs with signal-to-uncertainty ratios less than 100%. We assumed the retrieval uncertainty
of monthly mean OMI formaldehyde and glyoxal VCDs at $4^o \times 5^o$ resolution to be 40% and 100%,
respectively.

To remove globally systematic biases in the satellite observations, we adjusted the global observed
monthly mean VCDs by aligning the observed VCDs over remote reference areas to those simulated by
the GEOS-Chem model (sampled at satellite overpass time) using the *a priori* NMVOC emissions. The
remote Pacific ($140^o$-160W$^o$, 90$^o$S-90$^o$N) was chosen as the reference area for formaldehyde (Palmer et
al., 2003, 2006; Fu et al., 2007; González Abad et al., 2015). The Sahara desert (20-30$^o$N, 10$^o$W-30$^o$E),
where the interference from liquid water absorption was minimal, was chosen as the reference area for
glyoxal (Chan Miller et al., 2014). The justification for performing the alignment was two-fold: firstly,
the formaldehyde and glyoxal column concentrations over these remote reference areas were small and
well simulated by the model (Fu et al., 2008; Chan Miller et al., 2014). The removed biases over the
remote areas were less than 20% and 10% of the typical formaldehyde ($>8 \times 10^{15}$ molecule cm$^{-2}$) and
glyoxal ($>4 \times 10^{14}$ molecule cm$^{-2}$) monthly mean VCDs observed over eastern China, respectively.
More importantly, our inversion was performed over China only, assuming that the *a priori* NMVOC
emissions for the rest of the world were unbiased. As will be seen in Sections 3 and 4, the optimization
of NMVOC sources were predominantly driven by local formaldehyde and glyoxal enhancements
produced by relatively short-lived NMVOCs.

### 347    2.4 Inversion experiments using the GEOS-Chem adjoint

We used the GEOS-Chem model to perform Bayesian inversions on Chinese NMVOC emissions, using
satellite observations of formaldehyde and glyoxal over China and the *a priori* emission estimates as
constraints. The inversion minimized the cost function $J(\boldsymbol{x})$ in Eq. (1) (Rodgers, 2000), which we
calculated over China:
$$J\left(\boldsymbol{x}\right) = \gamma \cdot \left(\boldsymbol{x}-\boldsymbol{x_a}\right)\boldsymbol{S_a^{-1}}\left(\boldsymbol{x}-\boldsymbol{x_a}\right) + \left(\boldsymbol{F}\left(\boldsymbol{x}\right)-\boldsymbol{y}\right)\boldsymbol{S_o^{-1}}\left(\boldsymbol{F}\left(\boldsymbol{x}\right)-\boldsymbol{y}\right) \qquad \text{Eq. (1)}$$
The first and second terms on the right-hand-side of Eq. (1) represented the penalty error and the
prediction error, respectively. $\boldsymbol{x}$, which we sought to optimize, was the vector of scale factors (for each




NMVOC species from each emission sector) applied to the *a priori* emissions. $\mathbf{x_a}$ was a unit vector
applied to the *a priori* NMVOC emission estimates. $\mathbf{y}$ was the vector of satellite-observed monthly
mean VCDs of the targeted tracer (formaldehyde and/or glyoxal). $\mathbf{F(x)}$ was the vector of VCDs of the
targeted tracer simulated by the forward model $\mathbf{F}$. $\mathbf{S_a}$ was the *a priori* emission error covariance matrix,
which was a diagonal matrix with the uncertainties estimated based on ranges of previous NMVOC
estimates (Section 2.2 and Table 1).

The observation error covariance matrix in Eq. (1), $\mathbf{S_o}$, was difficult to quantify, as it included
contributions not only from the satellite retrieval, but also from the model representation of chemistry
and transport. Zhu et al. (2016) and Chan Miller et al. (2017) compared vertical profiles of
GEOS-Chem-simulated formaldehyde and glyoxal over the Southeast U.S. in summer against aircraft
measurements. They reported that the simulated formaldehyde mixing ratios showed only a small bias
(-3% ±2%) in the lower troposphere but were lower than the observations by 41% in the free
troposphere, likely due to insufficient deep convection in the model (Zhu et al., 2016). The simulated
glyoxal mixing ratios were within 20% of the observations in the mixed layer, but they were too low in
the upper troposphere by more than a factor of two, also likely due to insufficient model vertical
transport (Chan Miller et al., 2017). It should be noted that these errors assessed by Zhu et al. (2016)
and Chan Miller et al. (2017) likely also included the errors associated with precursor emissions.
Nevertheless, based on these assessments, we roughly estimated that the model errors for
formaldehyde and glyoxal VCDs to be ±80%, ±100%, respectively. Adding these estimated model
errors in quadrature to the satellite retrieval errors (Section 2.3), we estimated that the observation
error ($\mathbf{S_o}$) of formaldehyde and glyoxal to be about ±90% and ±150%, respectively.

The optimization of Eq. (1) was dependent on the relative weighting of the penalty error ($\mathbf{S_a}$) and the
prediction error ($\mathbf{S_o}$). However, the errors and error correlations within $\mathbf{S_a}$ and $\mathbf{S_o}$ were often
incompletely represented. In addition, we found that due to the mathematical formulation of Eq. (1),
the cost function $J(\mathbf{x})$ was heavily weighted by grids where the *a priori* estimates were too high, such
that the optimization was less effective at increasing emissions where the *a priori* emissions were too
low. These issues were empirically addressed in inversion studies by the introduction of a
regularization factor, γ, in Eq. (1) to adjust the relative weight of the penalty error. Henze et al. (2009)



used the L-curve method (Hansen, 1998) to find an optimized γ value, which minimized the total cost
function and balanced the prediction term and the penalty term. We followed that methodology and
found a γ value of 0.01 for July, which we applied for all of the warmer months (March to October). An
optimized γ value of 0.1 was found for January, and we applied that value to the colder months.

Table 2 shows the setup of our inversion experiments. We experimented with four different sets of
satellite retrievals as constraints, with the goal of bracketing the uncertainties of the top-down estimate
of Chinese NMVOC emissions. The first two experiments (IE-1 and IE-2) constrained emissions using
the formaldehyde and glyoxal VCDs observations from GOME-2A and OMI, respectively. Several
studies showed that GOME-2A formaldehyde VCDs may be low by a factor of 1.3 to 1.7 (Lee et al.,
2015; Zhu et al., 2016; Wang et al., 2017). As an "upper bound" constraint, we conducted a third
inversion experiment (IE-3), which was constrained by 1.7 times the GOME-2A formaldehyde alone.
We conducted a fourth inversion experiment (IE-4) constrained by OMI glyoxal VCDs alone to explore
the impacts of glyoxal observations on the inversions of anthropogenic emissions.

Figure 3 illustrates our protocol for the inversion experiments. For each month, we began by driving
the GEOS-Chem forward model with the *a priori* emissions ($x_{i=1} = x_a = 1$) to simulate the monthly mean
formaldehyde and glyoxal VCDs at satellite-crossing time. The simulated and satellite-observed VCDs
were used to calculate the cost function, $J(x)$, and the forcing arrays ($\frac{\partial J(x)}{\partial F(x)}$). The adjoint of
GEOS-Chem was then used to compute the cost function gradient ($\frac{\partial J(x)}{\partial x}$), and the next guess of the
emission scale factor ($x_{i+1}$) was calculated using the Quasi-Newton L-BFGS-B algorithm (Byrd et al.,
1995; Zhu et al., 1997), subject to the bounds $0.32 \leq x \leq 10$. These bounds were selected based on the
largest uncertainties quoted in the literature on Chinese NMVOC emission estimates (Q. Zhang et al.,
2009; Liu et al., 2012). The process was then iterated until the incremental relative reduction of the cost
function ($\frac{|J(x)_{i+1} - J(x)_i|}{\max(J(x)_{i+1}, J(x)_i)}$) was less than 1% after at least six iterations. We took $x_{i+1}$ from the last
iteration as the optimized emission scale factor ($x_p$) and applied it to calculate the top-down emission
estimate.



### 3  Comparison of simulations using the *a priori* emissions against satellite observations and ground-based measurements

We first qualitatively compared the formaldehyde and glyoxal VCDs simulated by the GEOS-Chem model (sampled at satellite overpass times) using the *a priori* emissions against those observed by GOME-2A and OMI, as well as ground-based measurements. Figure 4 (a)-(d) show the monthly mean formaldehyde VCDs observed by GOME-2A over China for January, April, June, and October 2007. Observed formaldehyde VCDs over China showed a distinct west-to-east gradient year-round, which was driven by the higher vegetation and population densities in eastern China. Observed formaldehyde VCDs were higher during the warmer months, reflecting the stronger biogenic emissions during the growing seasons. Highest formaldehyde VCDs were observed over the NCP in June, in response to the large emissions from in-field crop residue burning. In April, high concentrations of formaldehyde were also observed near the southwestern border, reflecting the seasonal biomass burning there.

Figure 4 (e)-(h) show the simulated monthly mean formaldehyde VCDs using the *a priori* emission estimates. The model generally reproduced the observed seasonal contrast and regional patterns. The simulated formaldehyde columns were higher than the GOME-2A observations in January, implying an overestimate of anthropogenic formaldehyde precursors in the *a priori* in January. The simulated formaldehyde columns were lower than the GOME-2A observations over eastern China in June, implying an underestimation of the emissions of formaldehyde precursors in June in the *a priori*.

A few ground-based measurements of tropospheric formaldehyde VCDs have been made in China using the MAX-DOAS technique (Li et al., 2013; Stavrakou et al., 2015; Wang et al., 2017) (Table S1). Figure 4 also shows the seasonal mean of these ground-based measurements at GOME-2A overpass time. In principle, these ground-based measurements were not directly comparable to the satellite-observed and model-simulated formaldehyde columns due to the different inherent uncertainties and the coarse spatial resolution of our analyses. Nevertheless, the seasonal progression presented by these few ground-based measurements were consistent with both the GOME-2A-observed and model-simulated formaldehyde VCDs.

Figure 5 shows the monthly mean glyoxal VCDs observed by GOME-2A. Similar to the case of





formaldehyde, GOME-2A-observed glyoxal VCDs were highest over eastern China in June, reflecting
large emissions of NMVOC species that are precursors to both formaldehyde and glyoxal. During
January the eastern China glyoxal enhancement was more evident than formaldehyde. As biogenic
emissions were small in winter, this indicated that the glyoxal VCDs were more reflective of
anthropogenic source. Figure 5 (e)-(h) shows that the simulated glyoxal VCDs were higher than the
GOME-2A observations in January and lower than the GOME-2A observations in June. This suggested
an overestimation of anthropogenic sources in January and an underestimation of the biogenic sources
in June, which was consistent with the constraints implied by the GOME-2A formaldehyde
observations. During the transition months of April and October, when the anthropogenic and biogenic
contributions to carbonyl productions were presumably more comparable, the simulated glyoxal VCDs
were lower than the GOME-2A observations, while the simulated formaldehyde VCDs were higher
than the GOME-2A observations (Figure 4 (e)-(h)). This likely indicated that the *a priori* inventory
underestimated the emissions of NMVOC species (e.g. aromatics, ethyne, ethene, and glyoxal) that
preferentially produced glyoxal, while it overestimated the emissions of species (e.g. $\geq C_4$ alkanes and
$\geq C_3$ alkenes from anthropogenic activities) that preferentially produced formaldehyde during the
transition months. Ground-based MAX-DOAS glyoxal measurements at a rural southern China site in
July 2006 averaged $6.8(\pm 5.2) \times 10^{14}$ molecules $cm^{-2}$, higher than both the GOME-2A-observed and
simulated glyoxal VCDs. No other ground-based measurements were available to provide spatial and
seasonal information.

Figure 6 (a)-(d) shows the monthly mean formaldehyde VCDs observed by the OMI instrument.
Similar to the GOME-2A-observed formaldehyde VCDs, OMI formaldehyde VCDs were higher over
Eastern China and enhanced during the warmer months. However, the formaldehyde VCDs observed
by OMI were lower than those observed by GOME-2A by approximately 30%, likely due to the
different retrieval algorithms (De Smedt et al., 2012; González Abad et al., 2015). The simulated
formaldehyde VCDs over China were also lower at OMI overpass time than at GOME-2A overpass
time by less than 20% in all seasons. However, the ground-based measurements at the three Chinese
surface sites did not consistently show such a diurnal pattern.





Figure 7 (a)-(d) shows the monthly mean glyoxal VCDs observed by the OMI instrument. Valid OMI
glyoxal VCDs observations were relatively sparse over China, especially during colder months. The
seasonal and spatial patterns of the glyoxal VCDs observed by OMI were generally consistent with
those observed by GOME-2A. However, the glyoxal VCDs observed by OMI were higher than those
observed by GOME-2A except in January. MAX-DOAS measurements of glyoxal at a rural southern
China site in July 2006 were also higher in the afternoon than in mid-morning. In contrast, the
simulated glyoxal VCDs at OMI overpass time (Figure 7 (e)-(h)) were lower than those at GOME-2A
overpass time. This discrepancy among the glyoxal diurnal cycles represented by the MAX-DOAS
measurements and the model indicated an uncertainty in the local glyoxal budget.

Figures 6 and 7 also compare the formaldehyde and glyoxal VCDs observed by OMI to those simulated
by the model using the *a priori* emission estimates over China. Formaldehyde VCDs observed by OMI
were lower than those simulated by the model in all seasons, with the exception of a local hotspot over
the NCP in June. However, the glyoxal VCDs observed by OMI were higher than those simulated by
the model in all seasons. It thus appeared that the constraints on Chinese NMVOC emissions indicated
by the OMI formaldehyde and glyoxal observations were contradictory, even during January and June
when the NMVOC emissions over Eastern China were dominated by anthropogenic and biogenic
sources, respectively. There may be two explanations for this apparent contradiction indicated by the
OMI formaldehyde and glyoxal observations. Firstly, the simulated photochemical budgets of
formaldehyde and glyoxal during the local afternoon may be in error. Errors in the model
photochemical budget would also explain why the MAX-DOAS measurements of formaldehyde and
glyoxal VCDs were both higher in the afternoon than in the morning, while the model showed an
opposite diurnal contrast. It is also possible that there were different inherent biases in the OMI
formaldehyde and glyoxal retrievals.

**4   Top-down estimates of Chinese NMVOC emissions**
**4.1   *A posteriori* formaldehyde and glyoxal VCDs from inversion experiments**
The qualitative analyses in Section 3 showed that the GOME-2A and OMI retrievals of formaldehyde
and glyoxal VCDs provide disparate information on seasonal Chinese NMVOC emissions. Thus, our



four inversion experiments on monthly Chinese NMVOC emissions using different satellite
observations as constraints (Table 2) represent a range of probable top-down estimates given current
satellite observations. Figure 2 shows the monthly top-down Chinese NMVOC emission estimates
from the four inversion experiments for January, April, June, and October and compares them against
the *a priori* emission estimates. The top-down emission estimates for the full twelve months are shown
in Figure S3. Figure S4 shows the changes in the normalized cost functions over China in the four
inversion experiments. Relative to their respective initial cost function values, the optimized cost
function values were reduced by 10%-60% for all four experiments.

Figure 4 (i-l) and Figure 5 (i-l) show the *a posteriori* monthly mean VCDs of formaldehyde and
glyoxal, respectively, from the GOME-2A formaldehyde-glyoxal inversion experiment IE-1. Overall,
IE-1 greatly improved the agreement between the *a posteriori* VCDs and the GOME-2A observations
for both formaldehyde and glyoxal. The *a posteriori* VCDs of formaldehyde and glyoxal over eastern
China both decreased in January and increased in June. During the transition months of April and
October, IE-1 decreased the *a posteriori* formaldehyde VCDs while increasing the *a posteriori* glyoxal
VCDs. Figure 2 illustrates how these changes in VCDs were driven by the top-down NMVOC
emission estimates. For IE-1, the estimated emissions of all NMVOC species were reduced in January
but enhanced in June. In April and October, however, IE-1 decreased the total NMVOC emissions
while preferentially increasing the emissions of anthropogenic glyoxal precursors.

Figure 6 (i-l) and Figure 7 (i-l) show the *a posteriori* monthly mean VCDs of formaldehyde and
glyoxal, respectively, from the OMI formaldehyde-glyoxal inversion experiment IE-2. IE-2 was
effective in reducing the *a posteriori* formaldehyde VCDs over eastern China year-round to better
agree with the OMI formaldehyde observations. However, the inversion increased the *a posteriori*
glyoxal VCDs only slightly and was less effective in bringing agreement with the OMI glyoxal
observations. Figure 2 shows that the *a posteriori* NMVOC emission estimates from IE-2 were lower
than the *a priori* estimates for all months. This was due to a combination of factors at work in the
inversion. The low formaldehyde observations from OMI in all months drove a large reduction in the
emissions of NMVOCs that produced only formaldehyde ($\geq C_4$ alkanes and $\geq C_3$ alkenes from



anthropogenic activities, as well as primary formaldehyde from biomass burning). At the same time,
the relatively high glyoxal observations from OMI drove an increase in the emissions of NMVOCs that
produced mainly glyoxal (ethene, ethyne, and aromatics from anthropogenic activities, as well as
primary glyoxal from biomass burning). For precursors that produced large amounts of both
formaldehyde and glyoxal (most importantly biogenic isoprene), the inversion reduced the top-down
emissions. This was because the formaldehyde observations had more leverage on the inversion due to
their lower observational errors. This manifested the importance of well-characterized retrievals with
reliable error estimates.

Figure 4 (q-t) shows the *a posteriori* formaldehyde VCDs from the inversion experiment IE-3, which
was constrained by the GOME-2A-observed formaldehyde VCDs scaled by a factor of 1.7. The *a*
*posteriori* formaldehyde VCDs in IE-3 were further increased during the warmer months relative to
IE-1, especially over the NCP and central China in June. In January, the scaled-up GOME-2A
observations were still lower than the simulated formaldehyde VCDs using the *a priori* emissions,
leading to a small reduction in the *a posteriori* formaldehyde VCDs. Figure 2 shows that the top-down
emission estimates of all NMVOC species were lower than the *a priori* in January and higher than the
*a priori* in June. Consequently, although no observations of glyoxal were used as constraints in IE-3,
the *a posteriori* glyoxal VCDs also decreased in January and increased in June (Figure 5(m) and (o)).
This is in agreement with our findings in Section 3, whereby the constraints exerted by the GOME-2A
formaldehyde and glyoxal observations were consistent in January and in June, when the NMVOC
emissions were dominated by anthropogenic and biogenic sources, respectively. However, IE-3 had
almost no effects on the simulated glyoxal VCDs and the top-down emission estimates of
anthropogenic glyoxal precursors in April and October (Figure 5 (n) and (p)). This demonstrated the
necessity of glyoxal observations on constraining the emissions of NMVOC species that preferentially
produced glyoxal, including most importantly aromatics.

The impacts of satellite glyoxal observations on constraining Chinese NMVOC emission estimates was
further demonstrated in IE-4. Figure 7 (m-p) shows the *a posteriori* glyoxal VCDs from IE-4, which
used only the OMI glyoxal observations as constraints. The *a posteriori* glyoxal VCDs for all months
increased, to an extent greater than those in IE-2. Figure 2 shows that this increase in the *a posteriori*



glyoxal VCDs in IE-4 was achieved mainly by increasing the emission estimates of anthropogenic
glyoxal precursors for all months. In June, the emissions of biogenic isoprene (precursor to both
glyoxal and formaldehyde) also increased. As a result, the *a posteriori* formaldehyde VCDs in IE-4
increased in June but remained similar to the *a priori* simulation for the other months (Figure 6 (m-p)).

**4.2 Top-down estimates of Chinese NMVOC emissions from inversion experiments**
Table 2 and Figure 8 shows the top-down estimates for Chinese annual total NMVOC emissions from
the four inversion experiments and compare them against the *a priori*. Our top-down annual total
estimates for Chinese NMVOCs ranged from 23.4 to 35.4 Tg C y$^{-1}$, compared to the 27.4 Tg C y$^{-1}$ of
the *a priori*. The highest top-down estimate was from IE-3, constrained by 1.7 times the GOME-2A
formaldehyde VCD observations. The lowest top-down estimate was from IE-2, due to the relatively
low formaldehyde observations from OMI.

Anthropogenic sources constituted 53%-57% of the total top-down NMVOC emissions. The highest
top-down anthropogenic emissions estimate was from IE-4 (19.7 Tg C y$^{-1}$), which reflected the strong
traction of the OMI glyoxal observations on constraining anthropogenic NMVOC emissions. The
lowest top-down anthropogenic emission estimate was from IE-2 (13.5 Tg C y$^{-1}$). All four inversion
experiments consistently showed larger emissions of anthropogenic glyoxal precursors than the *a priori*.
In particular, our top-down estimates for anthropogenic aromatics ranged from 5.0 to 7.3 Tg C y$^{-1}$,
consistently larger than the *a priori* of 4.9 Tg C y$^{-1}$ (Li et al., 2014).

The top-down estimates for biogenic NMVOCs emission ranged between 8.9 and 14.8 Tg C y$^{-1}$. The
top-down estimates for biogenic isoprene were 4.9 to 10.5 Tg C y$^{-1}$. The top-down estimate for biomass
burning NMVOC emissions were between 1.06 to 1.47 Tg C y$^{-1}$, with the largest top-down estimate
driven by the scaled-up GOME-2A formaldehyde VCDs (IE-3).

Figure 9 shows the spatial distribution of the scale factors for Chinese annual NMVOC emissions from
each of the four inversion experiments relative to the *a priori* emission estimates. The use of
GOME-2A formaldehyde and glyoxal observations as constraints in IE-1 led to a domain-wide increase



in biogenic NMVOC emissions, except in the northeast. IE-1 also found an increase in biomass burning
emissions over the NCP in June. A similar spatial distribution was found for the emission scale factors
of IE-3. Again, this indicated a consistency between the constraints exerted by the formaldehyde and
glyoxal observations from GOME-2A. The optimized emission scale factors from IE-2 and IE-4 were
of opposite signs. Using only OMI glyoxal observations as constraints in IE-4 led to a domain-wide
increase in NMVOC emissions from all sectors. However, when further constraints of the relatively
low OMI formaldehyde observations were added in IE-2, the top-down emission estimates decreased
across the domain.

As discussed previously, our four inversion experiments using different satellite retrievals as
constraints represent the range of probable top-down estimates given currently available satellite
observations. To represent the difference between these top-down estimates relative to the *a priori*, we
averaged the top-down estimates from the four inversion experiments. Our averaged top-down estimate
for Chinese total annual NMVOC emissions was 30.8 Tg C y$^{-1}$, including 17.0 Tg C y$^{-1}$, 12.6 Tg C y$^{-1}$,
and 1.2 Tg C y$^{-1}$ from anthropogenic, biogenic, and biomass burning sources, respectively. Our average
estimate for anthropogenic aromatic flux was 6.1 Tg C y$^{-1}$, which was 24% larger than the *a priori*
estimate of Li et al. (2014).

Figure 1 (e-l) shows the spatial distribution of annual Chinese NMVOC emission of our averaged
top-down estimate and the scale factors relative to the *a priori* estimates. Our averaged top-down
estimate of Chinese NMVOC emissions were spatially consistent with the *a priori*, but the total fluxes
increased by 10% to 40% throughout eastern China relative to the *a priori*. In particular, we found a 40%
increase in the biomass burning emissions over the NCP. We also found a 20%-40% increase in the
anthropogenic NMVOC emissions in coastal eastern China. Largest scale factors for biogenic NMVOC
were found near the northwestern border of China and along the northeast-to-southwest division line of
vegetation density. This potentially indicated an underestimation of biogenic NMVOC emission from
semi-arid ecosystems in the MEGAN inventory.



## 5    Comparison with previous estimates of Chinese NMVOC emissions

Table 1 compares our top-down estimates of Chinese annual NMVOC emissions for the year 2007 against estimates in the literature for the years between 2005 and 2012. It should be noted that bottom-up inventories often estimated the total NMVOCs emitted from a source sector using emission factors for total NMVOCs, then distributed the emissions using different species profile data. As a result, bottom-up estimates often included additional NMVOC species not represented here in our study.

Our top-down estimate for biogenic NMVOC emissions range from 8.9 to 14.8 Tg C $y^{-1}$, on average 17% larger than the flux calculated from the MEGAN inventory (Guenther et al., 2006). Our top-down estimate for isoprene emission (the single most emitted NMVOC species) ranged from 4.9 to 10.5 Tg C $y^{-1}$, bracketing the *a priori* of 6.6 Tg C $y^{-1}$. Stavrakou et al. (2015) previously used GOME-2A and OMI formaldehyde observations to derive top-down estimates of isoprene emissions over China of 5.0 Tg C $y^{-1}$ (GOME-2A) and 5.5 Tg C $y^{-1}$ (OMI), respectively. In comparison, our top-down isoprene emission estimates constrained by GOME-2A and OMI (both formaldehyde and glyoxal) observations, were 8.2 Tg C $y^{-1}$ (from IE-1) and 4.9 Tg C $y^{-1}$ (from IE-2), respectively. Our top-down estimates constrained by GOME-2A observations was larger than that of Stavrakou et al. (2015) due to the additional glyoxal constraints. Our estimate constrained by OMI observations was lower than that of Stavrakou et al. (2015) because the OMI formaldehyde VCDs over China retrieved by Gonz áez Abad et al. (2015) were systematically lower than the OMI formaldehyde VCDs retrieved by De Smedt et al. (2015).

Our top-down estimates for Chinese annual biomass burning NMVOC emissions ranged from 1.06 to 1.47 Tg C $y^{-1}$. These numbers are in good agreement with the estimate of Huang et al. (2012). Previous bottom-up biomass burning NMVOC emission estimates by Bo et al. (2008) and Wu et al. (2016) ranged from 1.9 to 2.4 Tg C $y^{-1}$, but only 25% to 30% of these emissions were from open burning of crop residues; the rest were emitted from biofuel burning. The GFED3 inventory (van der Werf et al., 2010), based on satellite burnt area observations, severely underestimated biomass burning emissions over China, particularly those associated with crop residue burning. Top-down estimate of Chinese biomass burning NMVOC emissions by Stavrakou et al. (2015) was between 1.1-1.5 Tg C $y^{-1}$, very





close to our top-down estimate range (1.06-1.47 Tg C y$^{-1}$). Similar to Fu et al. (2007) and Stavrakou et
al. (2016), our study also highlighted the large emissions from crop residue over the NCP in June
(Figure 2).

Previous bottom-up estimates of Chinese anthropogenic NMVOC emissions ranged widely from 10.7
to 29.8 Tg C y$^{-1}$ (Bo et al., 2008; Cao et al., 2011; Kurokawa et al., 2013; Li et al., 2014; Wu et al.,
2016) due to the use of different emission factors, activity data, and statistical models. Previous
top-down estimates of Chinese anthropogenic NMVOC emissions ranged from 17.3-28.7 Tg C y$^{-1}$ (Liu
et al., 2012; Stavrakou et al., 2015). Our top-down estimates had a smaller range between 13.5 to 19.7
Tg C y$^{-1}$. Our top-down estimates for anthropogenic aromatics (5.0 to 7.3 Tg C y$^{-1}$) were approximately
middle-of-the-range relative to previous estimates of 2.1-11.3 Tg C y$^{-1}$. The large difference between
previous top-down estimates and our top-down estimates of anthropogenic NMVOCs were
predominantly due to the choices of satellite observation constraints, and to a lesser extent due to the
choices of chemical transport model, the NMVOC species modeled, and the *a priori* emission
estimates. Specifically, the much higher estimate of anthropogenic aromatic emission by Liu et al.
(2012) (11.3 Tg C y$^{-1}$) compared to our top-down estimates (5.0-7.3 Tg C y$^{-1}$) was due to (1) the high
glyoxal VCDs observed by the SCIAMACHY instrument compared to those observed the GOME-2A
and OMI instruments over China; (2) the assumption made by Liu et al. (2012) that all anomalous
glyoxal was produced by aromatics oxidation; (3) the lower yields of glyoxal from aromatics oxidation
used in Liu et al. (2012) than those used in our model.

Our four inversion experiments all indicated stronger anthropogenic NMVOC emissions in summer
than in winter. In contrast, the *a priori* estimates showed a slightly stronger NMVOC emission in
winter than in summer, which was driven by stronger activity levels in winter in the bottom-up
inventories along with seasonally-invariant emission factors (Li et al., 2017). However, studies showed
that the NMVOC emission factors, in particular those for transport and industrial sectors, were strongly
and positively correlated with temperature (Rubin et al., 2006; Wei et al., 2016).



## 6 Impacts on simulated surface ozone levels over China

We evaluated the impacts on surface ozone concentrations due to our average top-down emission estimates of NMVOCs. Figure 10 compares the monthly mean afternoon (13:00-17:00 LT) surface ozone concentrations simulated using our averaged top-down emission estimates against those simulated using the *a priori* emissions for June and December 2007. Also shown in Figure 10 are surface observations at representative regional sites (Li et al., 2007; Xu et al., 2008; J. M. Zhang et al., 2009; Zheng et al., 2010; Wang et al., 2012; Wang et al., 2015; Li and Bian, 2015; Sun et al., 2016; Xu et al., 2016).Using the *a priori* emissions, the highest simulated afternoon surface ozone concentrations were between 100-110 ppb over the NCP in June. This was lower than the observations at two sites in the NCP, including a rural site outside Beijing (>100 ppb) and Mt. Tai (108 ppb). In comparison, by using our averaged top-down NMVOC emission estimate, the simulated afternoon surface ozone increased by 5-10 ppb over the NCP in June and were in better agreement with the observations. In December, the simulated afternoon surface ozone using *a priori* emission consistently overestimated the observed concentrations in eastern China. In comparison, by using our averaged top-down NMVOC emission estimates, the simulated afternoon surface ozone over eastern China decreased by 5 to 13 ppb, again in better agreement with the observations. It thus appears that our top-down emission estimates for Chinese NMVOCs improved simulation of regional ozone.

## 7 Conclusions

We used the GEOS-Chem model and its adjoint, as well as satellite observations of tropospheric column concentrations of formaldehyde and glyoxal, to constrain monthly Chinese NMVOC emissions from anthropogenic, biogenic, and biomass burning sources for the year 2007. We updated the gas-phase chemistry in the GEOS-Chem model and constructed its adjoint. The *a priori* emission estimates were taken from widely-used bottom-up emission inventories. We conducted four inversion experiments, which were constrained by formaldehyde and glyoxal observations from the GOME-2A instrument (IE-1), formaldehyde and glyoxal observations from the OMI instrument (IE-2), 1.7 times the formaldehyde observations from the GOME-2A instrument (IE-3), and glyoxal observations from the OMI instrument (IE-4), respectively. The results from these experiments represented the range of probable top-down NMVOC emission estimates for China given current satellite observational



constraints.

Our top-down estimates of total annual Chinese NMVOC emission from the four inversion
experiments ranged from 23.4 to 35.4 Tg C $y^{-1}$. Our top-down estimates of Chinese anthropogenic
NMVOC emission was 13.5 to 19.7 Tg C $y^{-1}$. In particular, we top-down estimate of Chinese
anthropogenic aromatic emissions range from 5.0 to 7.3 Tg C $y^{-1}$, much smaller than the top-down
estimate of 11.3 Tg C $y^{-1}$ by Liu et al. (2012). Our top-down estimate of Chinese biogenic NMVOC
emission ranged from 8.9 to 14.8 Tg C $y^{-1}$, with 4.9 to 10.5 Tg C $y^{-1}$ attributed to isoprene. Our
top-down estimate for Chinese biomass burning NMVOC emission range from 1.1 to 1.5 Tg C $y^{-1}$ and
was mostly associated with seasonal open burning of crop residue after local harvests, such as over the
NCP in June.

We evaluated the impacts on regional surface ozone concentrations from our average top-down
Chinese NMVOC emission estimates. We found that the simulated monthly mean afternoon surface
ozone concentrations increased by 5-12 ppb over the NCP in June, compared to the *a priori* simulation.
In December, the simulated monthly mean afternoon surface ozone concentrations decreased by 5-13
ppb over northern and central China, compared to the *a priori* simulation. For both seasons, the
simulation using our averaged top-down emission estimates were in better general agreement with
regional surface observations.

We concluded that formaldehyde and glyoxal observations from GOME-2A and OMI provide
quantitative constraints on the monthly emissions of Chinese NMVOCs. In particular, the simultaneous
use of the observations of both species helps distinguish NMVOC precursors and thus provides better
quantification of individual sources. However, better validation of these satellite data over China are
urgently needed, particularly in terms of discrepancies between different retrievals for the same species.

**Acknowledgements**
This work was supported by the Ministry of Science and Technology of China (2014CB441303) and
the National Natural Sciences Foundation of China (41461164007, 41222035). We thank National



Super Computer Center in Tianjin for supporting this work. D. K. Henze recognizes support from
NASA NNX17AF63G.

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

Table 1. Comparison of recent estimates for Chinese annual NMVOC emissions

| Literature [a] | Target year | NMVOC [Tg C y$^{-1}$] | | | | |
|---|---|---|---|---|---|---|
| | | Anthropogenic | | Biogenic | | Biomass burning |
| | | Total | Aromatics | Total | Isoprene | |
| Bottom-up | | | | | | |
| Bo et al. (2008) [b] | 2005 | 10.7 | | | | 2.2 [e] |
| Zhang et al. (2009) [b] | 2006 | 19.7 (±68%) | 2.1 | | | |
| Cao et al. (2011) [b] | 2007 | 29.8 | | | | |
| Kurokawa et al. (2013) [b] | 2008 | 22.8 (±46%) | | | | |
| Li et al. (2014) [b] | 2010 | 19.8 | 4.9 | | | |
| Wu et al. (2016) [b] | 2008 | 15.6 | | | | 2.2 [e] |
| | 2009 | 18.3 | | | | 1.9 [e] |
| | 2010 | 20.0 | | | | 2.1 [e] |
| | 2011 | 20.8 | | | | 2.1 [e] |
| | 2012 | 21.5 | | | | 2.4 [e] |
| Huang et al. (2012) [b] | 2006 | | | | | 1.3 (0.62-2.0) |
| van der Werf et al.(2010) | 2007 | | | | | 0.24 |
| Guenther et al.(2006) | 2007 | | | 10.8 [f] | 6.6 [f] | |
| Top-down | | | | | | |
| Liu et al. (2012) [c] | 2007 | 28.7 | 11.3 | | | |
| Stavrakou et al. (2015) [d] | 2010 | (17.3-20.7) | | | (5.0-5.5) | (1.1-1.5) |
| This work | 2007 | 17.0 [g] (13.5-19.7) | 6.1 [g] (5.0~7.3) | 12.6 [g] (8.9-14.8) | 8.4 [g] (4.9-10.5) | 1.22 [g] (1.06 – 1.47) |


[a] Emission estimates from literature were originally in units of Tg y$^{-1}$. We converted the units to Tg C y$^{-1}$ using to
carbon to organic compound mass ratios (0.84 for anthropogenic VOCs, 0.57 for biomass burning VOCs, and 0.85
for biogenic VOCs based on the *a priori* emission estimates).
[b] These emission estimates included NMVOC species that were not included in this work. See color keys in Figure
2 for NMVOC species whose emissions were included in this work.
[c] Used SCIAMACHY-observed glyoxal VCDs as constraints.
[d] Used GOME-2A-observed and OMI-observed formaldehyde VCDs as constraints.
[e] Consisted of emissions from open burning of crop residues and from biofuel burning.
[f] Calculated by the GEOS-Chem model using GEOS-5 meteorological data.
[g] Average of top-down estimates from four inversion experiments.











**Table 2. Inversion experiments to constrain Chinese NMVOC emissions**

| Inversion experiments | Observational constraints from satellites [±uncertainties] | Annual Chinese NMVOC emission estimates [Tg C y$^{-1}$] | | | |
|---|---|---|---|---|---|
| | | Anthropogenic | Biogenic | Biomass burning | Total |
| | | *A priori* emission estimates [±uncertainties] | | | |
| | | 15.5 (4.9 for aromatics) [a] [±200%] | 10.8 (6.6 for isoprene) [b] [±55%] | 1.10 [±300%] [c] | 27.4 |
| | | *A posteriori* emission estimates [range] | | | |
| IE-1 | GOME-2A formaldehyde [±90%] and glyoxal [±150%] | 15.7 (5.9 for aromatics) | 12.5 (8.2 for isoprene) | 1.13 | 29.3 |
| IE-2 | OMI formaldehyde [±90%] and glyoxal [±150%] | 13.5 (5.0 for aromatics) | 8.9 (4.9 for isoprene) | 1.06 | 23.4 |
| IE-3 | GOME-2A formaldehyde ×170% [±90%] | 19.2 (6.0 for aromatics) | 14.8 (10.5 for isoprene) | 1.47 | 35.4 |
| IE-4 | OMI glyoxal [±150%] | 19.7 (7.3 for aromatics) | 14.1 (9.9 for isoprene) | 1.24 | 35.1 |
| Our top-down estimates | | 17.0 [d] [13.5 - 19.7] (6.1 [d] [5.0 - 7.3] for aromatics) | 12.6 [d] [8.9 - 14.8] (8.4 [d] [4.9 – 10.5] for isoprene) | 1.2 [d] [1.1 - 1.5] | 30.8 [d] [23.4 – 35.4] |

[a] From Li et al. (2014)
[b] From Guenther et al. (2006).
[c] Compiled from the emission estimated by van der Werf et al. (2010) plus a scaling of the emission estimated by
Huang et al. (2012). See text (section 2.2) for details.
[d] Average of top-down estimates from the four inversion experiments.

















Table 3. Technical details for GOME-2A and OMI formaldehyde and glyoxal retrievals used in this study

| Technical details | | GOME-2A | | OMI | |
|---|---|---|---|---|---|
| | | Formaldehyde [a] | Glyoxal [b] | Formaldehyde [c] | Glyoxal [d] |
| Onboard satellite | | European Metop-A | | NASA Aura | |
| Operation time | | October 2006-present | | July 2004-present | |
| Overpass time | | 9:30 LT | | 13:30 LT | |
| Global coverage | | 1.5 days [e] | | 1 day | |
| Spatial resolution | | 80 km $\times$ 40 km | | 13 km $\times$ 24 km | |
| Spectral window | | 240-790 nm | | 270-500 nm | |
| Spectral resolution | | 0.26-0.5 nm | | 0.42 nm and 0.63 nm | |
| Selected absorption band | | 328.5 - 346 nm | 435 - 460 nm | 328.5 - 356.5 nm | 435 - 461 nm |
| Retrieval algorithm | | DOAS fitting | | Direct fitting | |
| Cloud parameters | | FRESCO+ (Wang et al., 2008) | | OMCLDO2 (Acarreta et al., 2004) | |
| Surface albedo | | Kleipool et al. (2008) | | Kleipool et al. (2008) | |
| Air mass factor calculation | Radiative transfer model | LIDORT (Spurr, 2008) | | VLIDORT (Spurr, 2006) | |
| | Tracer gas profiles | IMAGE v2 (Stavrakou et al., 2009b) | | GEOS-Chem (González Abad et al., 2015) | |
| Extinction by aerosols | | Considered implicitly via cloud correction | | Considered implicitly in the cloud retrieval | |
| Discarded pixels | | Cloud fraction > 40% or zenith angles >60° | | Cloud fraction > 40% | Impacted by random telegraph signals (RTS) [f] |

[a] From De Smedt et al. (2012)
[b] From Lerot et al. (2010)
[c] From González Abad et al. (2015)
[d] From Chan Miller et al. (2014)
[e] Before the swath was narrowed in June 2013. After that, the global coverage is achieved every 3 days.
[f] Pixels that have been flagged as RTS in the level 1-B product (Kleipool, 2005).





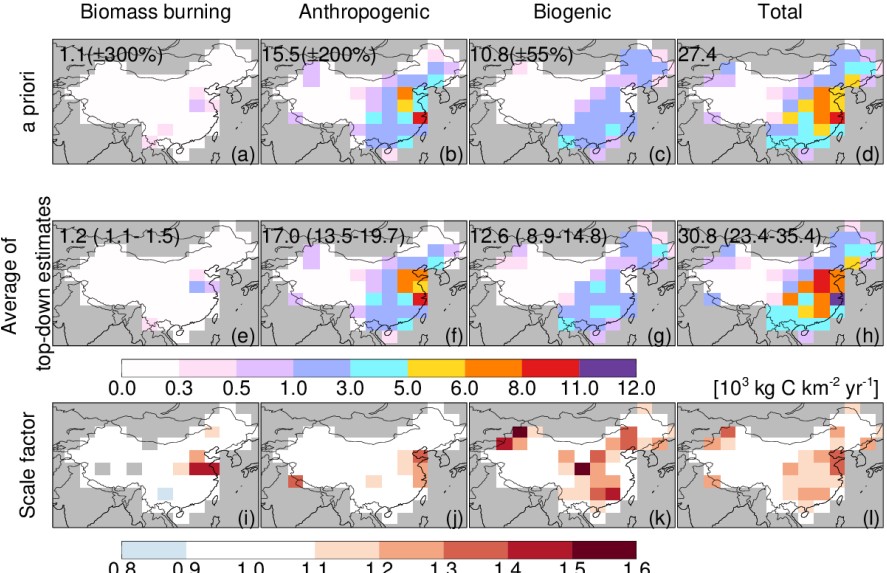

Figure 1. Spatial distributions of annual NMVOC emissions from China. (a)-(d): the *a priori* annual NMVOC emission estimates from (a) biomass burning, (b) anthropogenic, (c) biogenic, and (d) total sources. (e)-(h): averaged top-down estimates of annual NMVOC emissions. Annual Chinese total emission estimates are shown inset in units of [Tg C y$^{-1}$]. The uncertainties of the *a priori* emission estimates and the range of top-down emission estimates are shown in parentheses. (i)-(l): scale factors for our averaged top-down estimates relative to the *a priori* estimates.



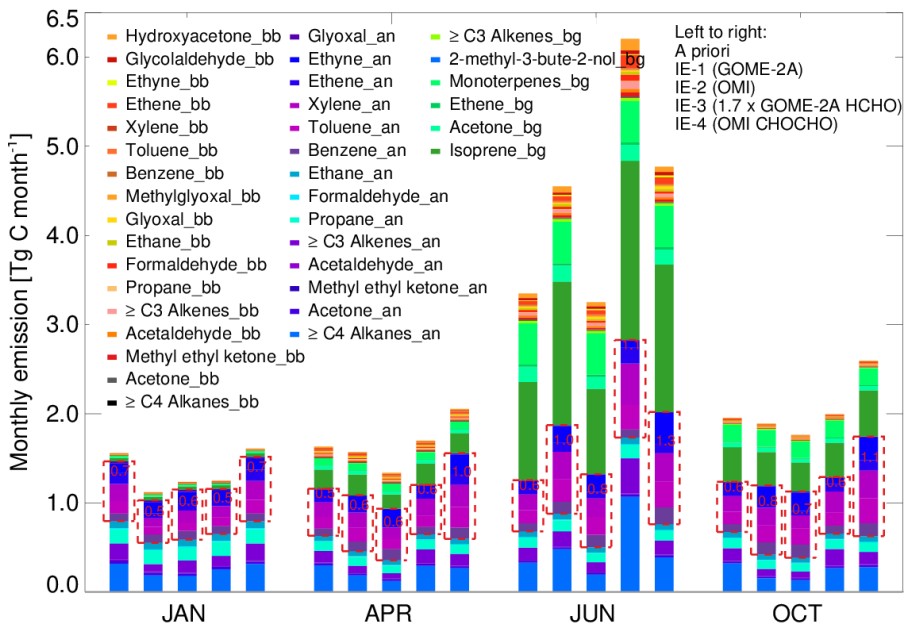

Figure 2. Estimates of monthly Chinese NMVOC emissions for January, April, June, and October 2007. For each month, the bars from left to right represent: the *a priori* emission estimates and the *a posteriori* emission estimates from IE-1, IE-2, IE-3, and IE-4. The red dashed boxes and red numbers indicate monthly emissions of anthropogenic glyoxal precursors. Color keys for NMVOC species are shown inset, with the suffixes of 'bb', 'an' and 'bg' indicating emissions from biomass burning, anthropogenic, and biogenic activities, respectively.

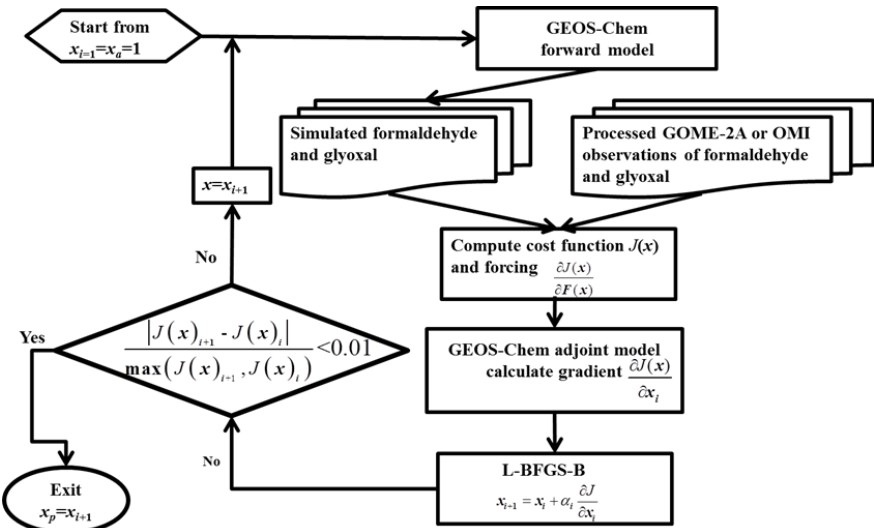

Figure 3. Protocol for the adjoint inversion experiments.



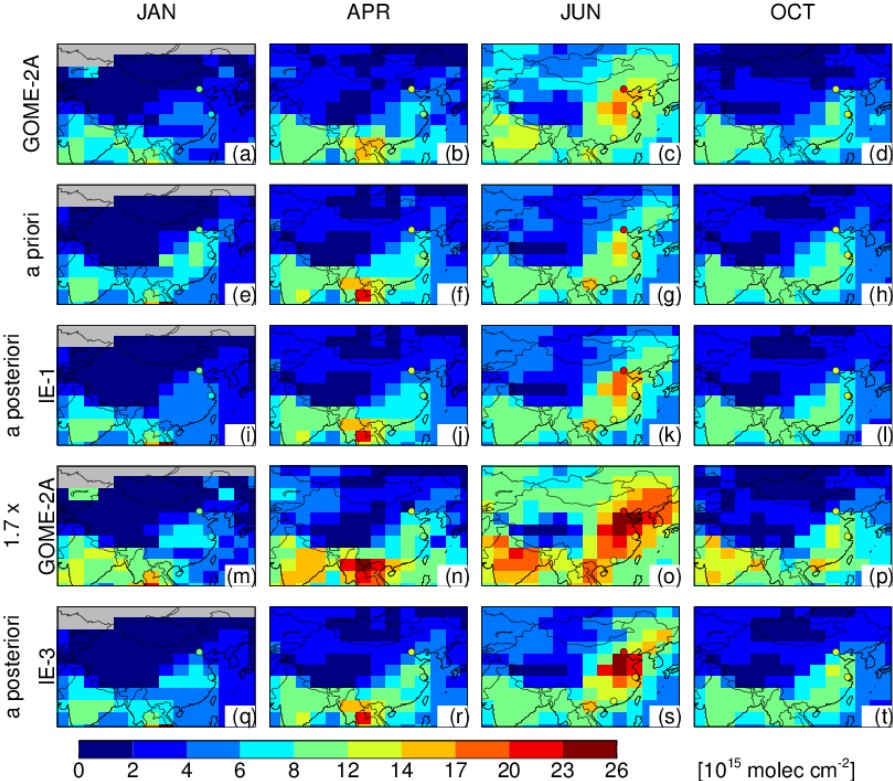

Figure 4. Monthly mean formaldehyde VCDs over China. (a-d): GOME-2A-observed formaldehyde VCDs and (m-p) GOME-2A formaldehyde VCDs scaled by a factor of 1.7. (e-h): Formaldehyde VCDs simulated by the model using *a priori* emission estimates; (i-l) the *a posteriori* formaldehyde VCDs from inversion IE-1;(q-t) the *a posteriori* formaldehyde VCDs from the inversion IE-3. Also shown are ground-based MAX-DOAS measurements at 9:30 LT (circles) at Beijing (De Smedt et al., 2015), Wuxi (Wang et al, 2017), and Back Garden (Li et al, 2013).



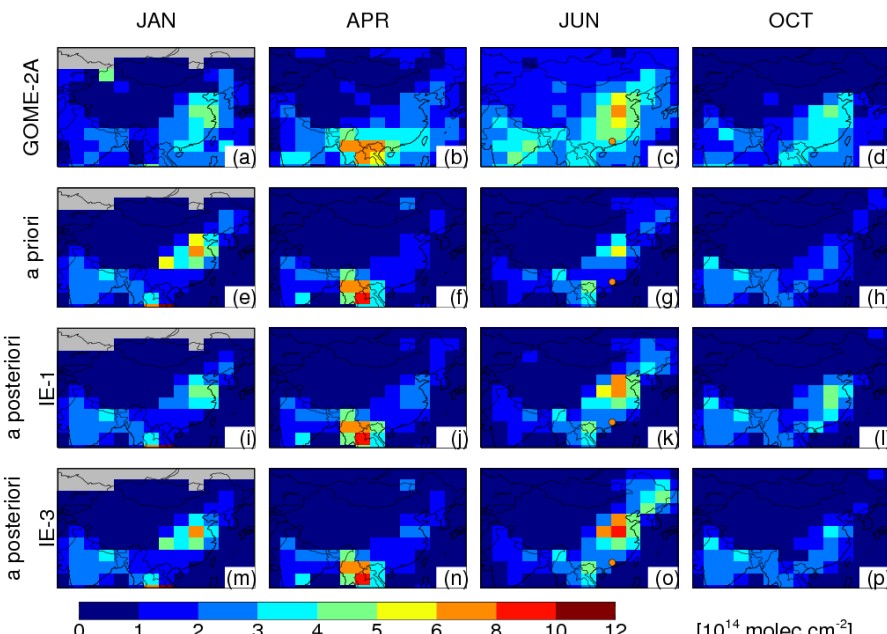

Figure 5. Monthly mean glyoxal VCDs over China (a-d) observed by the GOME-2A instrument, (e-h) simulated by the model using the *a priori* emission estimates, (i-l) obtain from inversion IE-1, and (m-p) obtain from inversion IE-3. Also shown are ground-based MAX-DOAS measurements at Back Garden in July 2006 (Li et al, 2013).





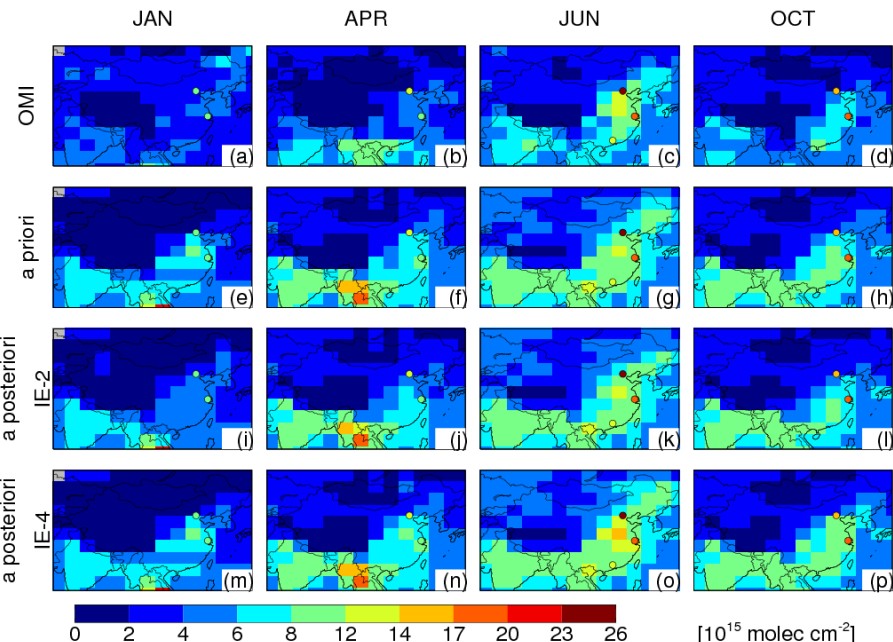

Figure 6. Monthly mean formaldehyde VCDs over China. (a-d): formaldehyde VCDs observed by the OMI instrument. (e-h): formaldehyde VCDs simulated by the model using the *a priori* emission estimates. (i-l): the *a posteriori* formaldehyde VCDs from inversion IE-2. (m-p): the *a posteriori* formaldehyde VCDs from inversion IE-4. Also shown are ground-based MAX-DOAS measurements at 13:30 LT (circles) at Beijing (De Smedt et al., 2015), Wuxi (Wang et al, 2017), and Back Garden (Li et al, 2013).





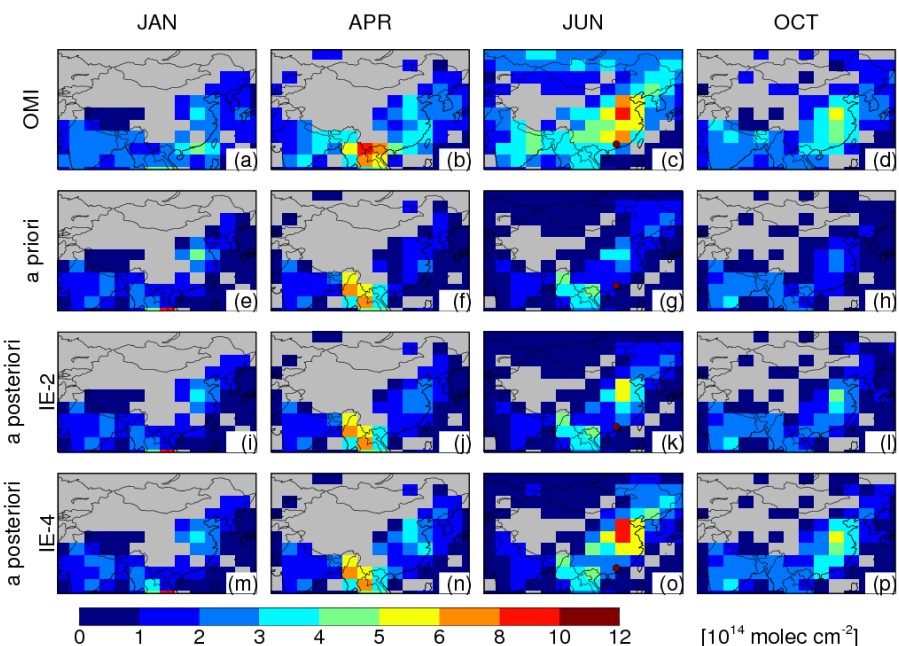

Figure 7. Monthly mean glyoxal VCDs over China (a-d) observed by the OMI instrument, (e-h) simulated by the model using the *a priori* emission estimates, (i-l) obtained from inversion IE-2, and (m-p) obtained from the inversion IE-4. Also shown are ground-based MAX-DOAS measurements at 13:30 LT (circles) at Back Garden in July 2006 (Li et al, 2013).




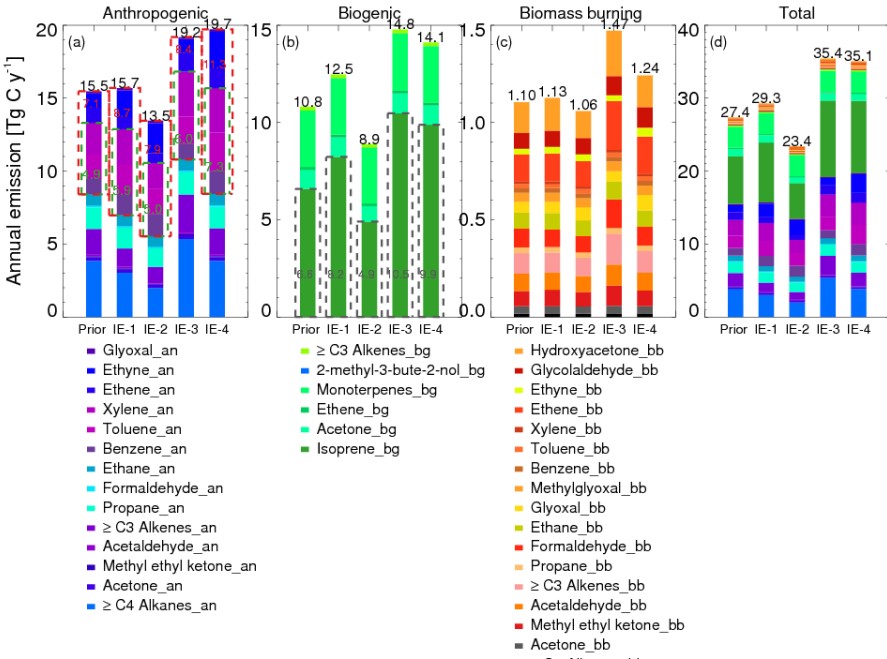

1190

Figure 8. Comparison of estimates of annual Chinese NMVOC emissions from (a) anthropogenic, (b) biogenic, (c) biomass burning, and (d) total sources. For each subfigure, shown from left to right are the *a priori* estimates and our *a posteriori* estimates from IE-1, IE-2, IE-3, and IE-4. Annual total NMVOC emission estimates are shown in black numbers on top of each bar. The red dashed boxes and red numbers in (a) indicate annual emissions of anthropogenic glyoxal precursors. The green dashed boxes and green numbers in (a) indicate annual emissions of anthropogenic aromatics. The grey dashed boxes and grey numbers in (b) indicate annual biogenic isoprene emissions. Color keys to NMVOC species are shown at the bottom, with suffixes of 'an', 'bg', 'bb' indicating anthropogenic source, biogenic source, and biomass burning source, respectively.






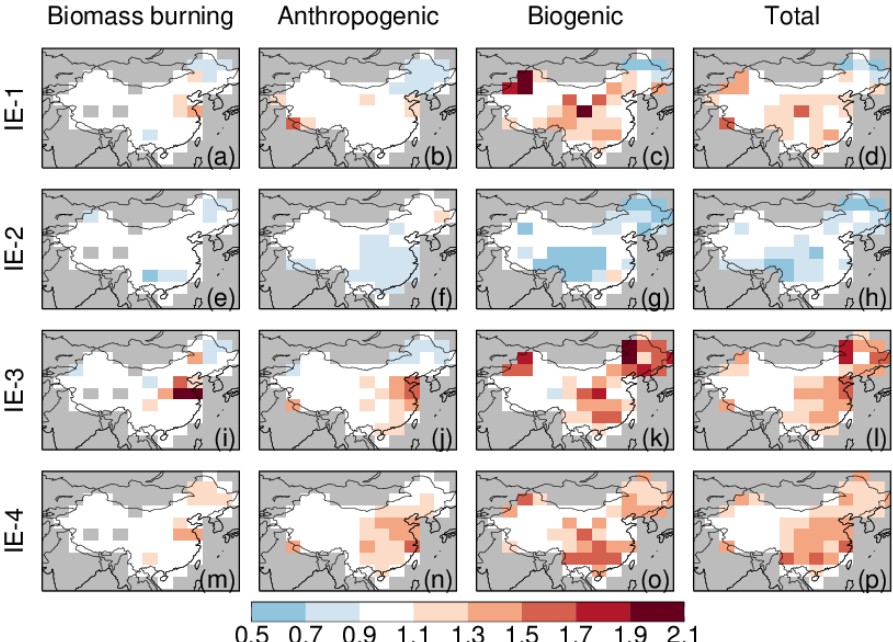


Figure 9. Spatial distributions of the optimized scale factors for Chinese annual NMVOC emissions, relative to the

*a priori* emission estimates, for the four inversion experiments.

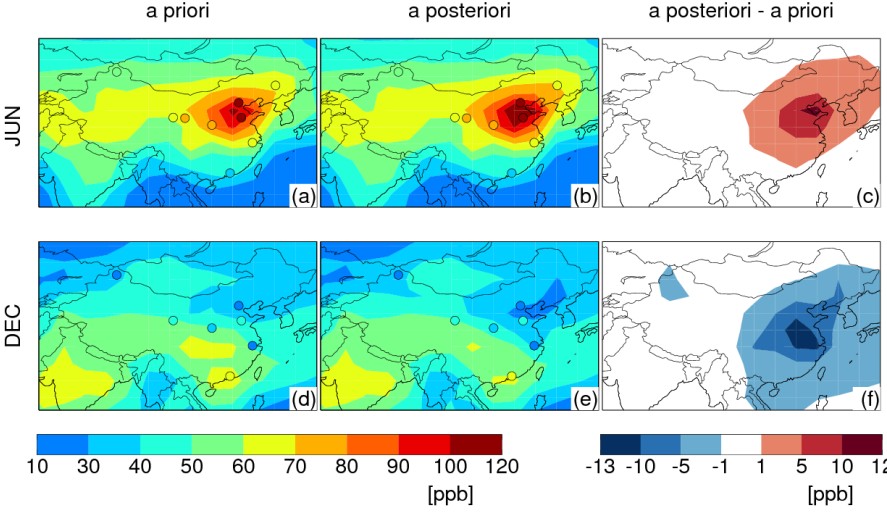


Figure 10. Simulated monthly mean afternoon (13:00-17:00 LT) surface ozone concentrations driven by the *a*
*priori* emissions and average of our top-down emissions, respectively, as well as corresponding difference (*a*
*posteriori–a priori*) in June and December 2007. Filled circles overlaid on the contour maps represent surface
ozone observations at several sites of China (Table S2).