# Peer review of "Adjoint inversion of Chinese non-methane volatile"

_Atmospheric Chemistry and Physics, 2017_

## Referee Comment (RC1) · Anonymous Referee #2 · 14 Feb 2018

**Cao et al., 2018, ACPD, Adjoint inversion of Chinese non-methane volatile organic compound emissions using space-based observations of formaldehyde and glyoxal**

**General Description of manuscript:**
The authors use satellite observations of glyoxal and formaldehyde to estimate a range in emissions of non-methane volatile organic compounds (NMVOCs) in China for 2007 using an adjoint inversion. Results from their inversion are discussed in the context of other top-down estimates for China and the a posteriori NMVOCs emissions are used to simulate surface ozone. The updated ozone concentrations increase consistency between the model and observed surface ozone concentrations in winter (December) and summer (July).

**General Comments:**
What are the implications of the updated NMVOCs emissions on organic aerosol (and hence PM$_{2.5}$) over China?

Why use such a coarse resolution version of GEOS-Chem (5x4), when higher resolution versions of GEOS-Chem are available for the globe (2.5x2) and nested over China (0.667x0.5 for GEOS-5 meteorology)?

What is the effect of the updates to the model (Section 2.1) on simulated column concentrations of HCHO and CHOCHO?

**Specific Comments:**
Lines 74-75: Include Millet et al. (2006) as a reference for the high yields of formaldehyde from NMVOCs.

Lines 77-80: Biogenic emissions doesn't always dominate HCHO columns over the Amazon and Africa. Both locations include a large and often dominant contribution from biomass burning to HCHO.

Lines 81-82: "linearly proportional to the local biogenic isoprene flux during the growing season" seems odd, in particular when the HCHO columns are used to estimate isoprene emissions. Do you mean that the vegetation distribution and HCHO are spatially correlated?

Line 80: Marais et al. (2012; 2014) obtained isoprene emissions for all of Africa, not just the tropical portion.

Line 84: The chronology is odd. The line starts with "Later studies", but many of these studies precede the studies in the previous paragraph.

Line 102: "diffused" should be "diffuse".

Line 117: Is "anonymous" a typo?

Lines 163-177: This paragraph needs more context for readers not familiar with the array of GEOS-Chem model versions and chemistry mechanisms. Is this a separate branch of the

model that includes detailed carbonyl chemistry not included in the standard version? What exactly are the updates that are applied to GEOS-Chem in this work? Has this branch of the model fallen behind the other model versions and so is being updated in this work to include the isoprene chemistry that is currently in the standard version of the model?

Line 171: Is v10-01 correct? The isoprene chemistry of Paulot et al. (2009a; b) was added to v9-02.

Line 181: Provide the yield values for Fu et al. (2008).

Line 183: Bloss et al. (2005) was used above as the reference for MCM v3.1. What is the appropriate reference for MCM v3.2?

Line 187: Does "our model" refer to GEOS-Chem?

Line 188: What was the Henry's law constant updated from and to?

Line 220: Specify which version of MEGAN is used in GEOS-Chem.

Liner 245: Was MEIC also scaled to 2007? As written this isn't clear.

Lines 250, 252, 643: "burnt" should be "burned".

Lines 253-254: Is the CO flux scaled or is CO used to estimate (or perhaps scale) NMVOC emissions?

Line 308: Should "IMAGE" by "IMAGES"?

Lines 445-447: The sentence beginning "As biogenic emissions…" is challenging to follow. Seems there's a logical step missing.

Line 464: "OMI formaldehyde VCDs were higher" than what? The a priori?

Lines 574-575: What does "strong traction" mean?

Figures 5 and 6: Are ground-based observations sampled at the same time as the satellite overpass?

Figures 4,6: "Monthly mean formaldehyde" in the figure caption is deceptive if seasonal means are shown for the ground-based observations.

Figures 4-7, 10: Increase the size of the points showing the ground-based measurements.

**References:**
Bloss et al., ACP, 2005, doi: 10.5194/acp-5-641-2005.
Chan Miller et al., ACP, 2016, doi:10.5194/acp-16-4631-2016.
Fu et al., JGR, 2008, doi:10.1029/2007jd009505.
Marais et al., ACP, 2012, doi:10.5194/acp-12-6219-2012.

Marais et al., ACP, 2014, doi:10.5194/acp-14-7693-2014.
Millet et al., JGR, 2006, doi:10.1029/2005jd006853.
Paulot et al., ACP, 2009a, doi:10.5194/acp-9-1479-2009.
Paulot et al., ACP, 2009b, doi: 10.1126/science.1172910.

---

## Referee Comment (RC2) · Anonymous Referee #1 · 9 Mar 2018

This study reports top-down estimates of non-methane volatime organic compound emissions over China based on formaldehyde and glyoxal column observations from two sounders, OMI and GOME-2 for 2007. Based on model simulations with the adjoint of the GEOS-Chem model, Cao et al. analyze the impacts of the different satellite datasets on the top-down emission estimates. They find that the annual total top-down VOC emission amounts to 30 Tg C, by 10% higher than the a priori inventory. In addition, using glyoxal retrievals from OMI, the authors estimate the annual aromatics Chinese source from 5 to 7.3 Tg C, also higher than in the bottom-up inventory. This study addresses an interesting subject for Atmospheric Chemistry and Physics journal. However, there are several weaknesses in the current work. For example, the figures

are not informative enough and cannot properly feed the discussion, the tables appear in an illogical order, some key statements appear without citation, references are missing. In addition, I see contradictions in the top-down emission estimates mentioned in the abstract and not enough details (and possibly errors) in the chemical scheme. Therefore, I have doubts regarding the validity of the conclusions and think that the manuscript will need a major revision before it becomes suitable for publication.

General comments :

- The chemical mechanism described very briefly in Section 2.1 is the core ingredient of the top-down VOC studies.

    – In l.164-165, several NMVOC precursors of formaldehyde are mentioned, but key precursors like methanol, acetaldehyde, ethanol, acetone, etc. do not show in the list. Why are these compounds omitted? Provide also more details on C4 alkanes (l.165).
    – In l.166 propane and (higher) alkanes are mentioned as glyoxal precursors. I have serious doubts on this. Please elaborate on the degradation scheme leading to glyoxal in your model.
    – l.170-172 : provide more details on how glyoxal is formed at both high- and low-NOx levels.
    – l.172-176 : I don't get this. Li et al. (2016) discusses the AM3 mechanism, not the GEOS-Chem mechanism. Furthermore, the statement that the updated scheme matches the MCM yields is not correct, the NOx-dependence of the yield is completely different in the two schemes.
    – Provide a table with formation yields at high and low NOx conditions for formaldehyde and glyoxal from their respective precursors.

- The comparisons between emission estimates shown in Table 1 relies heavily on conversion factors of 0.84, 0.57 and 0.85 for anthropogenic, biomass burning

and biogenic VOC, respectively. There is no reference on how these numbers are calculated. In particular, for isoprene and monoterpenes the factor of 0.85 is wrong. For methanol the real factor is also much lower.

- Table 1 misses emission estimates from widely used recent bottom-up and top-down inventories, e.g. GFED4 (van der Werf et al. 2017) on biomass burning emissions, HTAPV2 (Janssens-Maenhout et al. 2015) and EDGARv4.3.2 (Huang et al. 2017) on anthropogenic emissions, MEGAN-MACC (Sindelarova et al. 2014) and MEGAN-MOHYCAN (Stavrakou et al. 2014) on biogenic VOC, MACCity (Granier et al. 2011) on global anthropogenic and fire inventories. Especially for China, top-down estimates from Fu et al. (2008), Bauwens et al. (2016), Stavrakou et al. (2017), Granier et al. (2017) are missing.

- The Table ordering is illogical. Table 3 should rather become Table 1 or move to the supplement. Table 2 describes the simulations, so it should come first. Table 3 shows results and comparisons to previous studies so it should be called in the result section.

- In the abstract you mention that the annual total NMVOC emissions ranges from 23.5 to 35.4 Tg C (mean of 30.8). This does not match the sum of individual categories given in lines 27-29 of the abstract (23.5-36 Tg C). This brings confusion to the reader already from the first lines. Which one is correct? Change accordingly throughtout the paper and the Tables. In l.29-30 provide a name for the "most widely used boootm-up inventory".

- l.231 : Do you mean 19.8 Tg C from Table or am I missing something? I have several doubts about the reported numbers. Check again before you resubmit.

- In l. 239, the uncertainty of a priori emissions is given, $\pm 200\%$. Is this what is really meant here? It would correspond to a range of -20 to 60 Tg C. This makes

no sense given the reported numbers for the anthropogenic flux from different inventories. Same for l. 224, 267.

- l.249 : Liu et al. (2015) is based fire radiative power, not burnt area.

- l. 265 : GFED4 (van der Werf et al. 2017) acoounts for agricultural fire burning, which was not the case in GFED3. You should compare with GFED4 for this emission category.

- There are many language errors in the manuscript. This decreases its readability. I strongly recommend that the manuscript is corrected by a native speaker among the co-authors and thoroughly re-read.

- The discussion in Sections 3, 4 is not quantitative. The reader does not get enough information about absolute differences. This should be improved in the revised version.

- l. 466 : Can you specify what are the differences between the retrievals algorithms? I wonder why you didn't use retrievals from GOME-2 and OMI based on the same retrieval algorithm. These products are available. This should remove undesirable biases due to the different retrieval methodologies.

- All figures are based on model/data comparisons only for January, April, June, October. By doing that, we miss important information for other months, especially for July and August (maximum of biogenic emissions). The figures are also hard to read. More synthetic figures should be added, for instance showing the monthly variation of the satellite/model columns over large regions.

- Detailed comparisons with ground-based measurements are missing. The ground-based measurements shown in Figures 4-7 leave a lot to be desired. No concrete conclusion can be drawn from these plots with regards to the observed monthly variation and how well the model can reproduce it.

**Bibliography**

Fu, T.-M., D. J. Jacob, P. I. Palmer, K. Chance, Y. X. Wang, B. Barletta, D. R. Blake, J. C. Stanton, and M. J. Pilling, Space-based formaldehyde measurements as constraints on volatile organic compound emissions in East and South Asia and implications for ozone, J. Geophys. Res. 112 (D6), doi:10.1029/2006jd007853, 2008.

Granier, C., B. Bessagnet, T. Bond et al. Evolution of anthropogenic and biomass burning emissions of air pollutants at global and regional scales during the 1980-2010 period, Climatic Change, 109, 163–190, 2011.

Granier, C., T. Doumbia, L. Granier, K. Sindelarova, G. Frost, I. Bouarar, C. Liousse, S. Darras and J. Stavrakou: Anthropogenic emissions in Asia, Air Pollution in Eastern Asia : an integrated perspective, eds. Bouarar, I., Wang, X., Brasseur, G., Springer international Publishing, doi:10.1007/978-3-319-59489-7-6, pp. 107-133, 2017.

Huang, G., Brook, R., Crippa, M., Janssens-Maenhout, G., Schieberle, C., Dore, C., Guizzardi, D., Muntean, M., Schaaf, E., and Friedrich, R.: Speciation of anthropogenic emissions of non-methane volatile organic compounds: a global gridded data set for 1970-2012, Atmos. Chem. Phys., 17, 7683-7701, https://doi.org/10.5194/acp-17-7683-2017, 2017.

Janssens-Maenhout, G., Crippa, M., Guizzardi, D., Dentener, F., Muntean, M., Pouliot, G., Keating, T., Zhang, Q., Kurokawa, J., Wankmüller, R., Denier van der Gon, H., Kuenen, J. J. P., Klimont, Z., Frost, G., Darras, S., Koffi, B., and Li, M.: HTAPv2.2: a mosaic of regional and global emission grid maps for 2008 and 2010 to study hemispheric transport of air pollution, Atmos. Chem. Phys., 15, 11411-11432, https://doi.org/10.5194/acp-15-11411-2015, 2015.

Sindelarova, K., C. Granier, I. Bouarar, A. Guenther, S. Tilmes, T. Stavrakou, J.-F. Muller, U. Kuhn, P. Stefani, and W. Knorr, Global dataset of biogenic VOC emissions calculated by the MEGAN model over the last 30 years, Atmos. Chem. Phys., 14,
**Interactive comment**

9317-9341, 2014.

Stavrakou, T., J.-F. Muller, M. Bauwens, I. De Smedt, M. Van Roozendael, A. Guenther, M. Wild, and X. Xia, Isoprene emissions over Asia 1979-2012 : impact of climate and land use changes, Atmos. Chem. Phys., 14, 4587-4605, 2014.

Stavrakou, T., J.-F. Muller, M. Bauwens, I. De Smedt, Sources and long-term trends of ozone precursors to Asian Pollution, Air Pollution in Eastern Asia : an integrated perspective, eds. Bouarar, I., Wang, X., Brasseur, G., Springer international Publishing, doi:10.1007/978-3-319-59489-7-8, pp. 167-189, 2017.

van der Werf, G. R., Randerson, J. T., Giglio, L., van Leeuwen, T. T., Chen, Y., Rogers, B. M., Mu, M., van Marle, M. J. E., Morton, D. C., Collatz, G. J., Yokelson, R. J., and Kasibhatla, P. S.: Global fire emissions estimates during 1997-2016, Earth Syst. Sci. Data, 9, 697-720, https://doi.org/10.5194/essd-9-697-2017, 2017.
* * *

---

## Author Comment (AC1) · 30 Jun 2018

We thank the two reviewers for their constructive and detailed comments. In response, we have added additional analyses and re-written most of the main text to improve clarity throughout. We respond to each specific comment below. The reviewers' original comments are shown in red. Our replies are shown in black. The corresponding changes in the manuscript are shown in blue.
* * *
Reviewer 1:

General Description of manuscript:

The authors use satellite observations of glyoxal and formaldehyde to estimate a range in emissions of non-methane volatile organic compounds (NMVOCs) in China for 2007 using an adjoint inversion. Results from their inversion are discussed in the context of other top-down estimates for China and the a posteriori NMVOCs emissions are used to simulate surface ozone. The updated ozone concentrations increase consistency between the model and observed surface ozone concentrations in winter (December) and summer (July).

General Comments:

R1.1       What are the implications of the updated NMVOCs emissions on organic aerosol (and hence PM2.5) over China?

Thank you for the suggestion. We added in Section 6 an assessment of the impacts of our average top-down NMVOC emission estimates on simulated Chinese surface SOC (Figure S8), as well as comparison to surface SOC measurements (Table S10). We found that, by driving the model with our average top-down NMVOC emissions, the simulated surface SOC concentrations in June increased by 0.1 to 0.8 μgC m$^{-3}$ over eastern China relative to the simulation using the *a priori* NMVOC emissions. This increase in simulated SOC concentrations brought the model to closer to the surface measurements, but the model still severely underestimated observed SOC concentrations.

[Main text, lines 765 to 782]: Figure S8 compares the simulated monthly mean surface SOC concentrations using our averaged top-down NMVOCs emissions against those simulated using the *a priori* NMVOC emissions for January and June in 2007. Also shown are the SOC measurements at 12 surface sites in June of 2006 and 2007 from Zhang et al. (2012) (Table S10). By driving the model with our average top-down NMVOC emissions, the simulated surface SOC

concentrations in June increased by 0.1 to 0.8 µgC m⁻³ over eastern China relative to the simulation using the *a priori* NMVOC emissions. This increase in simulated SOC concentrations brought the model to closer to the surface measurements, but the model still severely underestimated observed SOC concentrations. We note our version of the GEOS-Chem model only included two pathways for secondary organic aerosol formation: (1) the reversible partitioning of semi-volatile products from the oxidation of isoprene, monoterpenes, and aromatics formation pathways (Liao et al., 2007; Henze et al., 2008), and (2) the irreversible uptake of dicarbonyl by aqueous aerosols and cloud drops (Fu et al., 2008). Other pathways, such as the atmospheric aging semi-volatile and intermediate volatility organic compounds (S/IVOC), has been shown to be an important source of secondary organic aerosols (Robinson et al., 2007; Pye and Seinfeld, 2010) but they were not included in our version of GEOS-Chem. In any case, the precursors and formation pathways of secondary organic aerosols in China are still poorly understood (Fu et al., 2012), such that no quantitative conclusions can be drawn regarding the impacts of our top-down NMVOC emission estimates on regional secondary organic aerosol formation.

[Supplementary information, Figure S8]:

[Figure]

**Figure S8. Simulated monthly mean surface secondary organic carbon (SOC) concentrations in June and December 2007 driven by (a) the *a priori* emissions and (b) our average top-down emissions, respectively, as well as (c) the differences. Overlaid symbols show the SOC measurements at 12 urban (circles) and regional (triangles) sites in China in June (Table S10). Mean biases (MB) of the simulated concentrations relative to surface measurements in June are shown inset.**

[Supplementary information, Table S10]: Surface measurements of SOC concentrations in June during 2006 and 2007 (Zhang et al., 2012)[a] and comparison to simulated SOC concentrations

| Site | Site type | SOC concentration (µg C m$^{-3}$) | | | Bias (model - measurement) | |
|---|---|---|---|---|---|---|
| | | measurement | *a priori* simulation | average top-down emission estimates simulation | *a priori* simulation | average top-down emission estimates simulation |
| Chengdu (30.65$^o$N, 104.03$^o$E) | urban | 3.79 | 1.31 | 1.61 | -2.49 | -2.18 |
| Dalian (38.9$^o$N, 121.63$^o$E) | urban | 2.64 | 1.32 | 2.09 | -1.32 | -0.55 |
| Dunhuang (40.15$^o$N, 94.68$^o$E) | regional | 2.51 | 0.38 | 0.41 | -2.13 | -2.11 |
| Gaolanshan (36.0$^o$N, 105.85$^o$E) | regional | 1.29 | 0.73 | 0.97 | -0.56 | -0.32 |
| Jinsha (29.63$^o$N, 114.2$^o$E) | regional | 1.81 | 1.40 | 1.85 | -0.42 | 0.03 |
| Lhasa (29.67$^o$N, 91.13$^o$E) | regional | 2.34 | 0.47 | 0.48 | -1.88 | -1.86 |
| LinAn (30.3$^o$N, 119.73$^o$E) | regional | 2.51 | 0.95 | 1.29 | -1.55 | -1.22 |
| Longfengshan (44.73$^o$N, 127.6$^o$E) | regional | 1.89 | 0.85 | 1.09 | -1.04 | -0.79 |
| Nanning (22.82$^o$N, 108.35$^o$E) | urban | 1.70 | 0.72 | 0.74 | -0.98 | -0.96 |
| Taiyangshan (29.17$^o$N, 111.71$^o$E) | regional | 1.11 | 1.38 | 1.72 | 0.27 | 0.61 |
| XiAn (34.43$^o$N, 108.97$^o$E) | urban | 5.41 | 1.70 | 2.39 | -3.71 | -3.02 |
| Zhengzhou | urban | 2.78 | 1.59 | 2.17 | -1.19 | -0.62 |

| | | | | | | |
|---|---|---|---|---|---|---|
| (34.78ºN, 113.68ºE) | | | | | | |
| Average | | 2.48 | 1.07 | 1.40 | -1.42 | -1.08 |

ª SOC concentrations were computed using organic carbon measurements (µgC m$^{-3}$) and the EC-tracer approach (Zhang et al., 2012).

**R1.2** Why use such a coarse resolution version of GEOS-Chem (5x4), when higher resolution versions of GEOS-Chem are available for the globe (2.5x2) and nested over China (0.667x0.5 for GEOS-5 meteorology)?

We agree with the Reviewer that our methodology is applicable to inversions at higher-resolutions and that it would be worthwhile to do so. However, the computation cost would be overwhelming for our analyses, which involved 48 inversion experiments (4 sets of satellite observations $\times$12 months, each inversion needed 10 to 50 calculations of forward and backward integrations) at higher resolutions (which would also require shorter time steps). We do plan to do higher resolution inversions focusing on a shorter periods of time, which would be more computationally feasible. We added a comment on this in the main text:

[Main text, lines 824 to 828]: The monthly inversions presented in this work, conducted at $5^{\circ}$ longitude $\times 4^{\circ}$ latitude resolutions due to limited computation resources, quantified the Chinese NMVOC emissions on regional/sub-regional scales. Future inversions and sensitivity studies targeting shorter periods of time may be conducted on finer resolutions to quantify Chinese NMVOC emissions and to evaluate their impacts on photochemistry at city cluster scales.

**R1.3** What is the effect of the updates to the model (Section 2.1) on simulated column concentrations of HCHO and CHOCHO?

Thank you for pointing out this lack of clarity. We added more detailed description of the updated chemical mechanisms, as well as a summary of the yields of formaldehyde and glyoxal from the oxidation of individual NMVOC precursors in our updated mechanisms (Table S1).

[Main text, lines 172 to 213]: We updated the dicarbonyl chemical mechanism in GEOS-Chem developed by Fu et al. (2008), which in turn was originally adapted from the Master Chemical Mechanism (MCM) version 3.1 (Jenkin et al., 1997; Saunders et al.,

2003). Table S1 lists the yields of formaldehyde and glyoxal from the OH-oxidation of NMVOC precursors in our updated chemical mechanism. The lumped NMVOC precursors of formaldehyde in our mechanism included ethane, propane, $\geq$C4 alkanes, ethene, $\geq$C3 alkenes, benzene, toluene, xylenes, isoprene, monoterpenes, acetone, hydroxyacetone, methygloxal, glycolaldehyde, acetaldehyde, 2-methyl-3-bute-nol, methy ethyl ketone, methanol, and ethanol (lumped into $\geq$C4 alkanes). The lumped NMVOC precursors of glyoxal in our mechanism included ethene, ethyne, benzene, toluene, xylenes, isoprene, monoterpenes, glycolaldehyde, and 2-methyl-3-bute-2-nol (MBO). Hereinafter we focused our discussion on these NMVOC precursors only, as their emissions may be constrained by formaldehyde and glyoxal observations.

The OH-oxidation of isoprene is a major source of both formaldehyde and glyoxal over China (Fu et al., 2007, 2008; Myriokefalitakis et al., 2008). We replaced the isoprene photochemical scheme with that used in GEOS-Chem v10.1, which included updates from Paulot et al. (2009a,b) and Mao et al. (2013). In this updated scheme, oxidation of isoprene by OH under high-$NO_x$ conditions produces formaldehyde and glyoxal at yields of 0.436 molecules per C and 0.0255 molecules per C, respectively (Table S1), mainly via the $RO_2$+NO pathways. Under low-NOx conditions, oxidation of isoprene by OH produces formaldehyde and glyoxal at yields of 0.38 molecules per C and 0.073 molecules per C, respectively (Table S1), via both $RO_2$+$HO_2$ and $RO_2$-isomerization reactions. Li et al. (2016) implemented this same isoprene photochemical scheme into a box model and compared the productions of formaldehyde and glyoxal from isoprene oxidation with those in the MCM version 3.3.1 (Jenkin et al., 2015). They showed that the production pathways and yields of formaldehyde and glyoxal were similar in the two schemes under the high-NOx conditions typical of eastern China.

We updated the molar yields of glyoxal from the OH oxidations of benzene (33.3%), toluene (26.2%), and xylenes (21.0%) following the latest literature (Arey et al., 2009; Nishino et al., 2010). These new molar yields were higher than those used in Fu et al. (2008) (based on averaged yields in the literature: 25.2% for benzene, 16.2% for toluene, and 15.6% for xylenes) but still lower than those used by Chan Miller et al. (2016) (75% for benzene, 70% for toluene, and 36% for xylenes), which were taken from the aromatic chemical scheme in MCM version 3.2 (Jenkin et al., 2003; Bloss et al., 2005). In MCM version 3.2, more than half of the glyoxal from aromatics oxidation were produced during second- and later-generation photochemistry, but such productions are with

limited experimental support and uncertain (Bloss et al., 2005).

Formaldehyde and glyoxal in the GEOS-Chem model were both removed by photolysis, as well as dry and wet deposition (Fu et al., 2008). We updated the Henry's law constant for glyoxal from $3.6 \times 10^5 \times \exp[7.2 \times 10^3 \times (1/T\text{-}1/298)]$ (Fu et al., 2008) to $4.19 \times 10^5 \times \exp[(62.2 \times 10^3/R) \times (1/T\text{-}1/298)]$ (Ip et al., 2009) and added the dry deposition of formaldehyde, glyoxal, methyglyoxal and glycolaldehyde on leaves (Mao et al., 2013). In addition, we assumed that glyoxal was reactively uptaken by wet aerosols and cloud droplets with an uptake coefficient = $2.9 \times 10^{-3}$ (Liggio et al., 2005; Fu et al., 2008). All other physical and chemical processes in our forward model were as described in Fu et al. (2008).

[Supplementary information, Table S1]:Ultimate yields of formaldehyde and glyoxal from the oxidation of NMVOC precursors by OH in our model under high-NOx and low-NOx conditions

| NMVOCs | Formaldehyde (molecules per C) | | Glyoxal (molecules per C) | |
|---|---|---|---|---|
| | High-NO$_x$ [a] | Low-NO$_x$ [b] | High-NO$_x$ [a] | Low-NO$_x$ [b] |
| Ethene | 0.995 | 0.366 | 0.0665 | 0.067 |
| Glycolaldehyde | 0.366 | 0.366 | 0.067 | 0.067 |
| Isoprene | 0.436 | 0.38 | 0.0255 | 0.073 |
| 2-methyl-3-bute-nol (MBO) | 0.092 | 0.092 | 0.0168 | 0.0168 |
| Benzene | 0.001 | 0.001 | 0.0555 | 0.0555 |
| Toluene | 0.198 | 0.18 | 0.037 | 0.037 |
| Xylenes | 0.269 | 0.155 | 0.026 | 0.026 |
| Monoterpenes (lumped) | 0.006 | 0.006 | 0.005 [c] | 0.005 [c] |
| Ethyne | - | - | 0.318 | 0.318 |
| Methanol | 1.0 | 1.0 | - | - |
| Ethane | 0.5 | 0.5 | - | - |
| Acetaldehyde (lumped) | 0.5 | 0.5 | - | - |
| Propane | 0.49 | 0.317 | - | - |
| $\geq$C$_3$ alkenes (lumped) | 0.657 | 0.333 | - | - |
| Acetone | 0.64 | 0.383 | - | - |
| Hydroxyacetone | 0.333 | 0.333 | - | - |
| Methyglyoxal | 0.333 | 0.333 | - | - |
| $\geq$C$_4$ alkanes (lumped) | 0.578 | 0.187 | - | - |
| Methy ethyl ketone (lumped) | 0.465 | 0.25 | - | - |

[a] Yields under high-NO$_x$ conditions were calculated assuming that all RO$_2$ radicals from the oxidation of the NMVOC precursor reacted with NO.

[b] Yields under low-NO$_x$ conditions were calculated assuming RO$_2$:HO$_2$ concentration ratio of 1:1.

[c] Glyoxal produced from the oxidation of monoterpenes by ozone

Specific Comments:

R1.4    Lines 74-75: Include Millet et al. (2006) as a reference for the high yields of formaldehyde from NMVOCs.

Added as suggested. Thank you.

> [Main text, lines 76 to 78]: Formaldehyde is produced at high yields during the oxidation of many NMVOC species (Millet et al., 2006) and also emitted directly from anthropogenic and biomass burning activities (Akagi et al., 2011; Li et al., 2017).

R1.5    Lines 77-80: Biogenic emissions doesn't always dominate HCHO columns over the Amazon and Africa. Both locations include a large and often dominant contribution from biomass burning to HCHO.

We revised this sentence as follows to avoid misunderstanding:

> [Main text, lines 78 to 83]: Early inversions of satellite-observed formaldehyde columns mostly focused on areas where the local NMVOC fluxes were dominated by biogenic sources during the growing season and in the absence of substantial biomass burning, such as the southeast U.S. (Palmer et al., 2003, 2006; Millet et al., 2006, 2008), Europe (Dufour et al., 2009; Curci et al., 2010), the Amazon (Barkley et al., 2008, 2009, 2013), and Africa (Marais et al., 2012, 2014a).

R1.6    Lines 81-82: "linearly proportional to the local biogenic isoprene flux during the growing season" seems odd, in particular when the HCHO columns are used to estimate isoprene emissions. Do you mean that the vegetation distribution and HCHO are spatially correlated?

We revised this sentence as follows to improve clarity:

[Main text, lines 83 to 84]: These studies showed that the observed local enhancements of formaldehyde column concentrations can be used to quantitatively constrain the local biogenic NMVOC fluxes.

R1.7     Line 80: Marais et al. (2012; 2014) obtained isoprene emissions for all of Africa, not just the tropical portion.

We re-wrote this sentence to correct for this error:

[Main text, lines 78 to 83]: Early inversions of satellite-observed formaldehyde columns mostly focused on areas where the local NMVOC fluxes were dominated by biogenic sources during the growing season and in the absence of substantial biomass burning, such as the southeast U.S. (Palmer et al., 2003, 2006; Millet et al., 2006, 2008), Europe (Dufour et al., 2009; Curci et al., 2010), the Amazon (Barkley et al., 2008, 2009, 2013), and Africa (Marais et al., 2012, 2014a).

R1.8     Line 84: The chronology is odd. The line starts with "Later studies", but many of these studies precede the studies in the previous paragraph.

Thank you for pointing this out. We re-wrote this sentences to improve clarity:

[Main text, lines 86 to 89]: In other areas, the NMVOC emissions from various sources may be comparable in magnitudes. Several studies constrained the NMVOC emissions from multiple sources over such areas by analyzing the spatiotemporal variability of the observed formaldehyde columns (Shim et al., 2005; Fu et al., 2007; Stavrakou et al., 2009b; Curci et al., 2010; Gonzi et al., 2011; Marais et al., 2014b; Zhu et al., 2014).

R1.9     Line 102: "diffused" should be "diffuse".

Corrected. Thank you.

R1.10     Line 117: Is "anonymous" a typo?

Yes, thank you for pointing out this typo. It should be "anomalous". We re-wrote the

sentence to avoid confusion:

[Main text line 120 to 123]: They suggested that the missing glyoxal source over eastern China was anthropogenic, on the basis that the anomalous glyoxal columns observed by SCIAMACHY (relative to the glyoxal columns simulated by their model) were spatially correlated with anthropogenic NOx emissions.

R1.11    Lines 163-177: This paragraph needs more context for readers not familiar with the array of GEOS-Chem model versions and chemistry mechanisms. Is this a separate branch of the model that includes detailed carbonyl chemistry not included in the standard version? What exactly are the updates that are applied to GEOS-Chem in this work? Has this branch of the model fallen behind the other model versions and so is being updated in this work to include the isoprene chemistry that is currently in the standard version of the model?

Thank you for pointing out this lack of clarity. We added the following description on the GEOS-Chem model version used in this work.

[Main text, lines 158 to 164]: We updated the GEOS-Chem global 3D chemical transport model (version 8.2.1) to simulate the emission, transport, chemistry, and deposition of NMVOCs, as well as the resulting formaldehyde and glyoxal column concentrations for the year 2007. The use of an older version of the GEOS-Chem forward model was necessary because, at the time of our study, the GEOS-Chem adjoint (version 34) was based on this older version. However, we updated the NMVOC chemical schemes (described below) and corrected several model errors in both our forward model and its adjoint by following the progress of the forward model up to version 10.1.

R1.12    Line 171: Is v10-01 correct? The isoprene chemistry of Paulot et al. (2009a; b) was added to v9-02.

Yes. The isoprene photochemical scheme in v10.1 included updates from Paulot et al. (2009a,b) and Mao et al (2013). We re-wrote this paragraph to clarify this point, as well as provide additional details on the updated isoprene photochemical scheme:

[Main text, lines 185 to 191]: We replaced the isoprene photochemical scheme with that used in GEOS-Chem v10.1, which included updates from Paulot et al. (2009a,b) and Mao et al. (2013). In this updated scheme, oxidation of isoprene by OH under high-$NO_x$

conditions produces formaldehyde and glyoxal at yields of 0.436 molecules per C and 0.0255 molecules per C, respectively (Table S1), mainly via the $RO_2+NO$ pathways. Under low-$NO_x$ conditions, oxidation of isoprene by OH produces formaldehyde and glyoxal at yields of 0.38 molecules per C and 0.073 molecules per C, respectively (Table S1), via both $RO_2+HO_2$ and $RO_2$-isomerization reactions.

R1.13   Line 181: Provide the yield values for Fu et al. (2008).

Thank you for the suggestion. We re-wrote this sentence to include the glyoxal yields from aromatics in Fu et al. (2008), those in our updated model, as well as those used by Chan Miller et al. (2016):

[Main text line 197 to 202]: We updated the molar yields of glyoxal from the OH oxidations of benzene (33.3%), toluene (26.2%), and xylenes (21.0%) following the latest literature (Arey et al., 2009; Nishino et al., 2010). These new molar yields were higher than those used in Fu et al. (2008) (based on averaged yields in the literature: 25.2% for benzene, 16.2% for toluene, and 15.6% for xylenes) but still lower than those used by Chan Miller et al. (2016) (75% for benzene, 70% for toluene, and 36% for xylenes), which were taken from the aromatic chemical scheme in MCM version 3.2 (Jenkin et al., 2003; Bloss et al., 2005).

R1.14   Line 183: Bloss et al. (2005) was used above as the reference for MCM v3.1. What is the appropriate reference for MCM v3.2?

Thank you for pointing out this error. We have updated the references for different updates to MCM:

[Main text, lines 172 to 174]: We updated the dicarbonyl chemical mechanism in GEOS-Chem developed by Fu et al. (2008), which in turn was originally adapted from the Master Chemical Mechanism (MCM) version 3.1 (Jenkin et al., 1997; Saunders et al., 2003).

[Main text, lines 198 to 205]: These new molar yields were higher than those used in Fu et al. (2008) (based on averaged yields in the literature: 25.2% for benzene, 16.2% for toluene, and 15.6% for xylenes) but still lower than those used by Chan Miller et al. (2016) (75% for benzene, 70% for toluene, and 36% for xylenes), which were taken from the aromatic chemical scheme in MCM version 3.2 (Jenkin et al., 2003; Bloss et al., 2005). In MCM version 3.2, more than half of the glyoxal from aromatics oxidation were produced during second- and later-generation photochemistry, but such productions are with limited

experimental support and uncertain (Bloss et al., 2005).

**R1.15**    Line 187: Does "our model" refer to GEOS-Chem?

Yes. Corrected to improve clarity:

[Main text, lines 207 to 208]: Formaldehyde and glyoxal in the GEOS-Chem model were both removed by photolysis, as well as dry and wet deposition (Fu et al., 2008).

**R1.16**    Line 188: What was the Henry's law constant updated from and to?

We added details about the updated Henry's law constant:

[Main text, lines 208 to 211]: We updated the Henry's law constant for glyoxal from $3.6 \times 10^5 \times \exp[7.2 \times 10^3 \times (1/T\text{-}1/298)]$ (Fu et al., 2008) to $4.19 \times 10^5 \times \exp[(62.2 \times 10^3/R) \times (1/T\text{-}1/298)]$ (Ip et al., 2009) and added the dry deposition of formaldehyde, glyoxal, methyglyoxal and glycolaldehyde on leaves (Mao et al., 2013).

**R1.17**    Line 220: Specify which version of MEGAN is used in GEOS-Chem.

We used MEGAN v2.0 (Guenther et al., 2006). This sentence was re-written as follows:

[Main text, lines 240 to 242]: The *a priori* biogenic NMVOC emissions from China and from the rest of the world were calculated with the MEGAN v2.0 algorithm (Guenther et al., 2006) and dependent on temperature, shortwave radiation, and monthly mean leaf area index.

**R1.18**    Liner 245: Was MEIC also scaled to 2007? As written this isn't clear.

No, we did not scale the MEIC emission estimates to the year 2007, because the uncertainty of the anthropogenic NMVOC emission estimates were much larger than the differences in emissions between the years 2007 and 2010. We added the following comment:

[Main text, lines 268 to 271]: As such, we did not scale the MEIC Chinese NMVOC emissions to the year 2007, because the uncertainty in the emission estimates were much larger than the differences in emissions between the years 2007 and 2010 (Chinese anthropogenic NMVOC emissions increased 14% from 2006 to 2010 according to Li et al, 2017).

R1.19    Lines 250, 252, 643: "burnt" should be "burned".

Corrected. Thank you.

R1.20    Lines 253-254: Is the CO flux scaled or is CO used to estimate (or perhaps scale) NMVOC emissions?

We used the CO emissions from crop residue burning estimated by Huang et al. (2012) and NMVOC-to-CO emission ratios for crop residue burning (Hays et al., 2002; Akagi et al., 2011) to estimate NMVOC emissions from crop residue burning. We rewrote the following sentences to make our treatment clear.

[Main text, lines 288 to 295]: Huang et al. (2012) estimated the Chinese CO emission from crop residue burning to be 4.0 Tg y$^{-1}$, based on MODIS daily thermal anomalies, Chinese provincial burned biomass data, and emission factors from Akagi et al. (2011). We scaled this CO flux using speciated NMVOC emission factors from crop residue burning from the literature (Hays et al., 2002; Akagi et al., 2011) and then multiplied the resulting NMVOC flux estimate by two. The reason for doubling the scaled NMVOC flux was that the emission factors for many NMVOC species were not measured, such that the sum of the speciated NMVOC emission factors was only half of the total NMVOC emission factor (Akagi et al., 2011).

R1.21    Line 308: Should "IMAGE" by "IMAGES"?

Corrected. Thank you.

R1.22    Lines 445-447: The sentence beginning "As biogenic emissions…" is challenging to follow. Seems there's a logical step missing.

We rewrote this sentence to improve clarity:

[Main text, lines 490 to 493]: During winter (particularly in January), the GOME-2A glyoxal VCDs show an enhancement over eastern China, which was not apparent in the GOME-2A formaldehyde VCDs. This indicated that the glyoxal VCDs were more reflective of anthropogenic source than formaldehyde VCDs.

R1.23    Line 464: "OMI formaldehyde VCDs were higher" than what? The *a priori*?

We rewrote this sentence to improve clarity:

[Main text, lines 513 to 515]: The spatial patterns and seasonal variations of the formaldehyde VCDs observed by OMI were similar to those observed by GOME-2A, with high formaldehyde over eastern China and during the warmer months.

R1.24    Lines 574-575: What does "strong traction" mean?

We rewrote this sentence to improve clarity:

[Main text, lines 590 to 593]: For precursors that produced large amounts of both formaldehyde and glyoxal (most importantly biogenic isoprene), the inversion reduced the top-down emissions as the formaldehyde observations had more weight in the cost function than the glyoxal observations, due to the lower observational errors in the formaldehyde VCDs.

R1.25    Figures 5 and 6: Are ground-based observations sampled at the same time as the satellite overpass?

Yes. We added clarification on this point in the main text, in the captions of Figures 3 to 10, and in the title of Table S3.

[Main text, lines 461 to 464]: A few ground-based measurements of tropospheric formaldehyde VCDs have been made in China using the Multi-Axis Differential Optical Absorption Spectrometry (MAX-DOAS) technique (Li et al., 2013; Vlemmix et al., 2015; Wang et al., 2017); these measurements (sampled at GOME-2A overpass time) are shown in Figure 3, Figure 4, and Table S3.

R1.26    Figures 4,6: "Monthly mean formaldehyde" in the figure caption is deceptive if seasonal means are shown for the ground-based observations.

Thank you for the suggestion. We now show all comparisons between satellite observations, model simulations, and ground-based MAX-DOAS measurements on a monthly basis, with the exception of measurements at Wuxi, which were only available as bi-monthly means. We added Table S3 to show the details of the MAX-DOAS measurements.

[Supplementary information, Table S3]: Ground-based MAX-DOAS measurements of formaldehyde and glyoxal vertical column densities in China at GOME-2A and OMI overpass times

| Reference | Location | Time of measurement | | Vertical column densities | |
|---|---|---|---|---|---|
| | | | | 9-10 local time | 13-14 local time |
| Formaldehyde [$10^{16}$ molecules cm$^{-2}$] | | | | | |
| Vlemmix et al. (2015) | Xianghe, Heibei (39.75N, 116.96E) | 2011 | JAN | 0.24 | 0.54 |
| | | | FEB | 0.78 | 0.99 |
| | | | MAR | 0.77 | 0.95 |
| | | | APR | 0.99 | 0.98 |
| | | | MAY | 1.08 | 1.53 |
| | | | JUN | 2.06 | 2.67 |
| | | | JUL | 1.49 | 2.10 |
| | | | AUG | 1.47 | 2.03 |
| | | | SEP | 1.05 | 1.36 |
| | | | OCT | 1.11 | 1.64 |
| | | | NOV | 0.85 | 1.18 |

| | | | | | |
|---|---|---|---|---|---|
| | | 2010 | DEC | 0.49 | 0.79 |
| Lee et al. (2015) | Beijing (39.59°N, 116.18°E) | August 16 to September 11, 2006 | | - | 1.79 |
| Wang et al. (2017) | Wuxi, Jiangsu (31.57°N,120.31°E) | 2011 - 2014 | JF | 0.7 [a] | 0.8 [a] |
| | | | MA | 0.9±0.15 [a] | 1.1±0.26 [a] |
| | | | MJ | 1.5±0.12 [a] | 1.9±0.15 [a] |
| | | | JA | 1.7±0.10 [a] | 2.2±0.26 [a] |
| | | | SO | 1.2±0.12 [a] | 1.7±0.12 [a] |
| | | | ND | 0.8±0.30 [a] | 1.4±0.32 [a] |
| Li et al. (2013) | Back Garden, Guangdong (23.50°N, 113.03°E) | July 2006 | | 1.3±1.0 [b] | 1.3±0.7 [b] |
| Glyoxal [$10^{14}$ molecules cm$^{-2}$] | | | | | |
| Li et al. (2013) | Back Garden, Guangdong (23.50°N, 113.03°E) | July 2006 | | 6.8±5.2 [c] | 11.4±6.8 [c] |

[a] From Figure 12 of Wang et al. (2017)

[b] From Figure 4 of Li et al. (2013)

[c] From Figure 5 of Li et al. (2013)

R.1.27    Figures 4-7, 10: Increase the size of the points showing the ground-based measurements.

The symbols in Figures 3 to 10, Figure 13, and Figure S8 have been enlarged as recommended. Thank you.
* * *
Reviewer 2:

This study reports top-down estimates of non-methane volatime organic compound emissions over China based on formaldehyde and glyoxal column observations from two sounders, OMI and GOME-2 for 2007. Based on model simulations with the adjoint of the GEOS-Chem model, Cao et al. analyze the impacts of the different satellite datasets on the top-down emission estimates. They find that the annual total top-down VOC emission amounts to 30 Tg C, by 10% higher than the *a priori* inventory. In addition, using glyoxal retrievals from OMI, the authors estimate the annual aromatics Chinese source from 5 to 7.3 Tg C, also higher than in the bottom-up inventory. This study addresses an interesting subject for Atmospheric Chemistry and Physics journal. However, there are several weaknesses in the current work.

For example, the figures are not informative enough and cannot properly feed the discussion, the tables appear in an illogical order, some key statements appear without citation, references are missing. In addition, I see contradictions in the top-down emission estimates mentioned in the abstract and not enough details (and possibly errors) in the chemical scheme. Therefore, I have doubts regarding the validity of the conclusions and think that the manuscript will need a major revision before it becomes suitable for publication.

General comments :

R2.1    The chemical mechanism described very briefly in Section 2.1 is the core ingredient of the top-down VOC studies.

In l.164-165, several NMVOC precursors of formaldehyde are mentioned, but key precursors like methanol, acetaldehyde, ethanol, acetone, etc. do not show in the list. Why are these compounds omitted? Provide also more details on C4 alkanes (l.165).

Thank you for pointing out this lack of clarity. We made major revisions to our mechanism and now included methanol as an independent tracer. Anthropogenic ethanol was lumped into $\geqslant C_4$ alkanes. Chinese biogenic ethanol was not included due to its small source. We rewrote the description of our NMVOC precursors to formaldehyde and glyoxal to improve clarity:

[Main text, lines 175 to 182]: The lumped NMVOC precursors of formaldehyde in our mechanism included ethane, propane, $\geqslant$C4 alkanes, ethene, $\geqslant$C3 alkenes, benzene, toluene, xylenes, isoprene, monoterpenes, acetone, hydroxyacetone, methygloxal, glycolaldehyde, acetaldehyde, 2-methyl-3-bute-nol, methy ethyl ketone, methanol, and ethanol (lumped into $\geqslant$C4 alkanes). The lumped NMVOC precursors of glyoxal in our mechanism included ethene, ethyne, benzene, toluene, xylenes, isoprene, monoterpenes, glycolaldehyde, and 2-methyl-3-bute-2-nol (MBO). Hereinafter we focused our discussion on these NMVOC precursors only, as their emissions may be constrained by formaldehyde and glyoxal observations.

R2.2    In l.166 propane and (higher) alkanes are mentioned as glyoxal precursors. I have serious doubts on this. Please elaborate on the degradation scheme leading to glyoxal in your model.

Thank you for pointing this out. Propane and high alkanes were not glyoxal precursors in our model. The original statement was a typo on our part, which has been removed. We

rewrote the description of our NMVOC precursors to formaldehyde and glyoxal to improve clarity:

[Main text, lines 175 to 182]: The lumped NMVOC precursors of formaldehyde in our mechanism included ethane, propane, ≥C4 alkanes, ethene, ≥C3 alkenes, benzene, toluene, xylenes, isoprene, monoterpenes, acetone, hydroxyacetone, methygloxal, glycolaldehyde, acetaldehyde, 2-methyl-3-bute-nol, methy ethyl ketone, methanol, and ethanol (lumped into ≥C4 alkanes). The lumped NMVOC precursors of glyoxal in our mechanism included ethene, ethyne, benzene, toluene, xylenes, isoprene, monoterpenes, glycolaldehyde, and 2-methyl-3-bute-2-nol (MBO). Hereinafter we focused our discussion on these NMVOC precursors only, as their emissions may be constrained by formaldehyde and glyoxal observations.

R2.3    l.170-172 : provide more details on how glyoxal is formed at both high- and low-NOx levels.

Thank you for the suggestion. We added more details on the formation of glyoxal from isoprene oxidation in the main text, as well as a summary of the yields of formaldehyde and glyoxal from individual NMVOC precursors (Table S1).

[Main text, lines 187 to 195]: In this updated scheme, oxidation of isoprene by OH under high-NOx conditions produces formaldehyde and glyoxal at yields of 0.436 molecules per C and 0.0255 molecules per C, respectively (Table S1), mainly via the $RO_2$+NO pathways. Under low-$NO_x$ conditions, oxidation of isoprene by OH produces formaldehyde and glyoxal at yields of 0.38 molecules per C and 0.073 molecules per C, respectively (Table S1), via both $RO_2$+$HO_2$ and $RO_2$-isomerization reactions. Li et al. (2016) implemented this same isoprene photochemical scheme into a box model and compared the productions of formaldehyde and glyoxal from isoprene oxidation with those in the MCM version 3.3.1 (Jenkin et al., 2015). They showed that the production pathways and yields of formaldehyde and glyoxal were similar in the two schemes under the high-NOx conditions typical of eastern China.

R2.4    l.172-176 : I don't get this. Li et al. (2016) discusses the AM3 mechanism, not the GEOS-Chem mechanism. Furthermore, the statement that the updated scheme matches the MCM yields is not correct, the NOx-dependence of the yield is completely different in the two schemes.

Thank you for pointing out this lack of clarity. The isoprene photochemistry mechanism which Li et al. (2016) implemented into the AM3 model was from GEOS-Chem v10.1 and

identical to the one we used. The Reviewer was correct in that the NOx dependence of the glyoxal yield from isoprene oxidation were different between the GEOS-Chem v10.1 mechanism (which we used) and the MCM v3.3.1. Our point was that, under the high-NOx conditions typical of eastern China, the production pathways and yields of formaldehyde and glyoxal were similar in these two mechanisms.

We rewrote this paragraph to improve clarity:

[Main text, lines 184 to 195]: The OH-oxidation of isoprene is a major source of both formaldehyde and glyoxal over China (Fu et al., 2007, 2008; Myriokefalitakis et al., 2008). We replaced the isoprene photochemical scheme with that used in GEOS-Chem v10.1, which included updates from Paulot et al. (2009a,b) and Mao et al. (2013). In this updated scheme, oxidation of isoprene by OH under high-$NO_x$ conditions produces formaldehyde and glyoxal at yields of 0.436 molecules per C and 0.0255 molecules per C, respectively (Table S1), mainly via the $RO_2$+NO pathways. Under low-$NO_x$ conditions, oxidation of isoprene by OH produces formaldehyde and glyoxal at yields of 0.38 molecules per C and 0.073 molecules per C, respectively (Table S1), via both $RO_2$+$HO_2$ and $RO_2$-isomerization reactions. Li et al. (2016) implemented this same isoprene photochemical scheme into a box model and compared the productions of formaldehyde and glyoxal from isoprene oxidation with those in the MCM version 3.3.1 (Jenkin et al., 2015). They showed that the production pathways and yields of formaldehyde and glyoxal were similar in the two schemes under the high-$NO_x$ conditions typical of eastern China.

R2.5    Provide a table with formation yields at high and low NOx conditions for formaldehyde and glyoxal from their respective precursors.

Thank you for this suggestion. We added Table S1 to summarize the yields of formaldehyde and glyoxal from the oxidation of individual NMVOC precursors under high- and low-$NO_x$ conditions.

[Supplementary information, Table S1]: Ultimate yields of formaldehyde and glyoxal from the oxidation of NMVOC precursors by OH in our model under high-NOx and low-NOx conditions

| NMVOCs | Formaldehyde (molecules per C) | | Glyoxal (molecules per C) | |
|---|---|---|---|---|
| | High-$NO_x$ [a] | Low-$NO_x$ [b] | High-$NO_x$ [a] | Low-$NO_x$ [b] |

| | | | | |
|---|---|---|---|---|
| Ethene | 0.995 | 0.366 | 0.0665 | 0.067 |
| Glycolaldehyde | 0.366 | 0.366 | 0.067 | 0.067 |
| Isoprene | 0.436 | 0.38 | 0.0255 | 0.073 |
| 2-methyl-3-bute-nol (MBO) | 0.092 | 0.092 | 0.0168 | 0.0168 |
| Benzene | 0.001 | 0.001 | 0.0555 | 0.0555 |
| Toluene | 0.198 | 0.18 | 0.037 | 0.037 |
| Xylenes | 0.269 | 0.155 | 0.026 | 0.026 |
| Monoterpenes (lumped) | 0.006 | 0.006 | 0.005 [c] | 0.005 [c] |
| Ethyne | - | - | 0.318 | 0.318 |
| Methanol | 1.0 | 1.0 | - | - |
| Ethane | 0.5 | 0.5 | - | - |
| Acetaldehyde (lumped) | 0.5 | 0.5 | - | - |
| Propane | 0.49 | 0.317 | - | - |
| $\geq C_3$ alkenes (lumped) | 0.657 | 0.333 | - | - |
| Acetone | 0.64 | 0.383 | - | - |
| Hydroxyacetone | 0.333 | 0.333 | - | - |
| Methyglyoxal | 0.333 | 0.333 | - | - |
| $\geq C_4$ alkanes (lumped) | 0.578 | 0.187 | - | - |
| Methy ethyl ketone (lumped) | 0.465 | 0.25 | - | - |

[a] Yields under high-$NO_x$ conditions were calculated assuming that all $RO_2$ radicals from the oxidation of the NMVOC precursor reacted with NO.

[b] Yields under low-$NO_x$ conditions were calculated assuming $RO_2$:$HO_2$ concentration ratio of 1:1.

[c] Glyoxal produced from the oxidation of monoterpenes by ozone

R2.6    The comparisons between emission estimates shown in Table 1 relies heavily on conversion factors of 0.84, 0.57 and 0.85 for anthropogenic, biomass burning C2 and biogenic VOC, respectively. There is no reference on how these numbers are calculated. In particular, for isoprene and monoterpenes the factor of 0.85 is wrong. For methanol the real factor is also much lower.

Thank you for pointing out this lack of clarity. In response, we have changed the unit for NMVOC emissions from Tg C y$^{-1}$ to Tg y$^{-1}$ to avoid the use of NMVOC carbon conversion factors.

R2.7    Table 1 misses emission estimates from widely used recent bottom-up and topdown inventories, e.g. GFED4 (van der Werf et al. 2017) on biomass burning emissions, HTAPV2 (Janssens-Maenhout et al. 2015) and EDGARv4.3.2 (Huang et al. 2017) on anthropogenic emissions, MEGAN-MACC (Sindelarova et al. 2014) and MEGAN-MOHYCAN (Stavrakou

et al. 2014) on biogenic VOC, MACCity (Granier et al. 2011) on global anthropogenic and fire inventories. Especially for China, top-down estimates from Fu et al. (2008), Bauwens et al. (2016), Stavrakou et al. (2017), Granier et al. (2017) are missing.

Thank you for this suggestion. We added most of these additional emission estimates to Table 2. The HTAPv2 emission estimates (Janssens-Maenhout et al., 2015) was not included in Table 2, as they were actually the MEIC emission estimates from Li et al. (2017).

[Main text, Table 2]: Comparison of Chinese annual NMVOC emission estimates for the years 2000 to 2014

| Literature | Target year | NMVOC [Tg y⁻¹] | | | | |
|---|---|---|---|---|---|---|
| | | Anthropogenic | | Biogenic | | Biomass |
| | | Total | Aromatics | Total | Isoprene | burning |
| *Bottom-up estimates* | | | | | | |
| Bo et al. (2008) [a] | 2005 | 12.7 | | | | 3.8 [d] |
| Zhang et al. (2009) [a] | 2006 | 23.2 (±68%) | 2.4 | | | |
| Cao et al. (2011) [a] | 2007 | 35.46 | | | | |
| Huang et al. (2017) [a] | 2007 | 24.6 | | | | |
| Granier et al. (2017) [a] | 2007 | 29.0 | | | | |
| Kurokawa et al. (2013) [a] | 2008 | 27.1 (±46%) | | | | |
| Li et al. (2017) [a] | 2010 | 23.6 | 5.4 | | | |
| Wu et al. (2016) [a] | 2008 | 18.62 | | | | 3.83 [d] |
| | 2009 | 21.8 | | | | 3.32 [d] |
| | 2010 | 23.83 | | | | 3.75 [d] |
| | 2011 | 24.78 | | | | 3.76 [d] |
| | 2012 | 25.65 | | | | 4.20 [d] |
| Huang et al. (2012) [a] | 2006 | | | | | 2.2 (1.08 to 3.46) |
| van der Werf et al. (2010) | 2007 | | | | | 0.47 |
| van der Werf et al. (2017) [a] | 2007 | | | | | 0.91 |
| Sindelarova et al. (2014) | 2005 | | | | 9.9 | |
| Guenther et al.(2006) | 2007 | | | 17.3 [e] | 7.5 [e] | |
| Stavrakou et al. (2014) | 2007 | | | | 7.6 | |
| *Top-down estimates* | | | | | | |
| Fu et al. (2007) | 2000 | 4.27 [g] | | 12.7 | | 5.1 |
| Liu et al. (2012) [b] | 2007 | 34.2 | 13.4 | | | |
| Stavrakou et al. (2014) | 2007 | | | | 8.6 | |
| Stavrakou et al. (2015) [c] | 2010 | 20.6 to 24.6 | | | 5.9 to 6.5 | 2.0 to 2.7 |
| Stavrakou et al. (2017) [c] | 2005 | 24.4 | | | 5.8 | |

| | | | | | | |
|---|---|---|---|---|---|---|
| | 2006 | 24.0 | | | (average | |
| | 2007 | 26.7 | | | of | |
| | 2008 | 25.9 | | | emissions | |
| | 2009 | 26.5 | | | from 2005 | |
| | 2010 | 26.1 | | | to 2011) | |
| | 2011 | 25.5 | | | | |
| | 2012 | 25.6 | | | | |
| | 2013 | 27.7 | | | | |
| | 2014 | 27.8 | | | | |
| This work | 2007 | 20.2 [f] (16.4 - 23.6) | 6.5 [f] (5.5 - 7.9) | 19.2 [f] (12.2 - 22.8) | 9.6 [f] (5.4 - 11.7) | 2.48 [f] (2.08 – 3.13) |

[a] These emission estimates included some NMVOC species which were not precursors to formaldehyde or glyoxal and therefore not included in this work. See color keys in Figure 2 for NMVOC species whose emissions were included in this work.

[b] Used SCIAMACHY-observed glyoxal VCDs as constraints.

[c] Used GOME-2A-observed and OMI-observed formaldehyde VCDs as constraints.

[d] Consisted of emissions from open burning of crop residues and from biofuel burning.

[e] Calculated by the GEOS-Chem model using GEOS-5 meteorological data.

[f] Average of top-down estimates from four inversion experiments.

[g] Only anthropogenic emissions of reactive alkenes, formaldehyde, and xylenes from northeastern, northern, central and southern China were included

R2.8    The Table ordering is illogical. Table 3 should rather become Table 1 or move to the supplement. Table 2 describes the simulations, so it should come first. Table 3 shows results and comparisons to previous studies so it should be called in the result section.

Thank you for the suggestion. We re-ordered the presentation of the tables in the main text as follows:

[Main text, Table 1]: Inversion experiments to constrain Chinese NMVOC emissions

[revised manuscript text omitted]

R2.9    In the abstract you mention that the annual total NMVOC emissions ranges from 23.5 to 35.4 Tg C (mean of 30.8). This does not match the sum of individual categories given in lines 27-29 of the abstract (23.5-36 Tg C). This brings confusion to the reader already from the first lines. Which one is correct? Change accordingly throughtout the paper and the Tables. In l.29-30 provide a name for the "most widely used bootom-up inventory".

Thank you for point out this lack of clarity, which was originally due to the expression of NMVOC emissions in units of Tg C y$^{-1}$. We have changed the unit for NMVOC emissions from Tg C y$^{-1}$ to Tg y$^{-1}$ to make the numbers consistent. We also re-wrote the description of the *a priori* emission inventories.

[Main text, Abstract, line 27 to 30]: Our top-down estimates for Chinese annual total NMVOC emission were 30.7 to 49.5 (average 41.9) Tg y$^{-1}$, including 16.4 to 23.6 (average 20.2) Tg y$^{-1}$ from anthropogenic sources, 12.2 to 22.8 (average 19.2) Tg y$^{-1}$ from biogenic sources, and 2.08 to 3.13 (average 2.48) Tg y$^{-1}$ from biomass burning.

[Main text, lines 798 to 806]: Our top-down estimates of total annual Chinese NMVOC emission from the four inversion experiments ranged from 30.7 to 49.5 Tg y$^{-1}$. Our top-down estimates of Chinese anthropogenic NMVOC emission was 16.4 to 23.6 Tg y$^{-1}$. In particular, our top-down estimates for Chinese anthropogenic aromatic emissions ranged from 5.5 to 7.9 Tg y$^{-1}$, much smaller than the top-down estimate of 13.4 Tg y$^{-1}$ by Liu et al. (2012). Our top-down estimate of Chinese biogenic NMVOC emission ranged from 12.2 to 22.8 Tg y$^{-1}$, with 5.4 to 11.7 Tg y$^{-1}$ attributed to isoprene. Our top-down estimate for Chinese biomass burning NMVOC emission range from 2.08 to 3.13 Tg y$^{-1}$ and was mostly associated with seasonal open burning of crop residue after local harvests, such as those over the NCP in June.

[Main text, lines 788 to 791]: The a priori NMVOC emission estimates from biogenic, anthropogenic, and biomass burning sources were taken from the inventories developed by Guenther et al. (2006), Li et al (2014, 2017), and Huang et al. (2012), as well as van der Werf et al. (2010), respectively.

R2.10    l.231 : Do you mean 19.8 Tg C from Table or am I missing something? I have several doubts about the reported numbers. Check again before you resubmit.

Please see the response to the previous comment.

R2.11    In l. 239, the uncertainty of *a priori* emissions is given, ±200%. Is this what is really meant here? It would correspond to a range of -20 to 60 Tg C. This makes no sense given the reported numbers for the anthropogenic flux from different inventories. Same for l. 224, 267.

Thank you for pointing out this lack of clarity. We re-wrote the statements on the uncertainty of the *a priori* emission estimates to avoid confusion and to maintain consistency with the original descriptions in the paper by Li et al. (2017).

[Main text line 266 to line 268]: We therefore estimated the uncertainty for the a priori Chinese anthropogenic NMVOC emission estimates to be a factor of two.

[Main text line 305 to line 307]: We therefore estimated the uncertainty of the a priori Chinese biomass burning NMVOC flux to be a factor of three.

R2.12    l.249 : Liu et al. (2015) is based fire radiative power, not burnt area.

Thank you for pointing out this lack of clarity. The original sentence meant that Liu et al. (2015) pointed out the underestimation of emissions in inventories based on satellite burned area observations. We re-wrote this paragraph to improve clarity:

[Main text, lines 281 to 288]: Post-harvest, in-field burning of crop residue has been recognized as a large seasonal source of NMVOCs in China (Fu et al., 2007; Huang et al., 2012; Liu et al., 2015; Stavrakou et al., 2016). These emissions from crop residue fires have been severely underestimated in inventories based on burned area observations from satellites, such as the Global Fire Emissions Database version 3 (GFED3, van der Werf et al., 2010). The recent Global Fire Emissions Database version 4 (GFED4, van der Werf et al., 2017) included small fires by scaling burned area with satellite fire pixel observations, but the resulting Chinese NMVOC emission estimate from biomass burning (0.91 Tg y$^{-1}$) was still much lower than the bottom-up inventory by Huang et al. (2012).

R2.13    l. 265 : GFED4 (van der Werf et al. 2017) acoounts for agricultural fire burning, which was not the case in GFED3. You should compare with GFED4 for this emission category.

Thank you for the suggestion. However, the NMVOC emissions from small fires in GFED4 were still much lower than both the estimates by Huang et al. (2012) and our top-down estimates. We added these comparisons in the main text and in Table 2.

[Main text, lines 285 to 288]: The recent Global Fire Emissions Database version 4 (GFED4, van der Werf et al., 2017) included small fires by scaling burned area with satellite fire pixel observations, but the resulting Chinese NMVOC emission estimate from biomass burning (0.91 Tg y$^{-1}$) was still much lower than the bottom-up inventory by Huang et al. (2012).

[Main text, lines 715 to 718]: The updated GFED4 (van der Werf et al., 2017) partially accounted for emissions for small fires, but its estimate for Chinese biomass burning NMVOC emissions was still lower than our top-down estimates by at least a factor of two.

[Main text, Table 2]: Comparison of Chinese annual NMVOC emission estimates for the years 2000 to 2014

| Literature | Target year | NMVOC [Tg y$^{-1}$] | | | | |
|---|---|---|---|---|---|---|
| | | Anthropogenic | | Biogenic | | Biomass |
| | | Total | Aromatics | Total | Isoprene | burning |
| *Bottom-up estimates* | | | | | | |
| Bo et al. (2008) [a] | 2005 | 12.7 | | | | 3.8 [d] |
| Zhang et al. (2009) [a] | 2006 | 23.2 (±68%) | 2.4 | | | |
| Cao et al. (2011) [a] | 2007 | 35.46 | | | | |
| Huang et al. (2017) [a] | 2007 | 24.6 | | | | |
| Granier et al. (2017) [a] | 2007 | 29.0 | | | | |
| Kurokawa et al. (2013) [a] | 2008 | 27.1 (±46%) | | | | |
| Li et al. (2017) [a] | 2010 | 23.6 | 5.4 | | | |
| Wu et al. (2016) [a] | 2008 | 18.62 | | | | 3.83 [d] |
| | 2009 | 21.8 | | | | 3.32 [d] |
| | 2010 | 23.83 | | | | 3.75 [d] |
| | 2011 | 24.78 | | | | 3.76 [d] |
| | 2012 | 25.65 | | | | 4.20 [d] |
| Huang et al. (2012) [a] | 2006 | | | | | 2.2 (1.08 to 3.46) |
| van der Werf et al. (2010) | 2007 | | | | | 0.47 |
| van der Werf et al. (2017) [a] | 2007 | | | | | 0.91 |
| Sindelarova et al. (2014) | 2005 | | | | 9.9 | |
| Guenther et al.(2006) | 2007 | | | 17.3 [e] | 7.5 [e] | |
| Stavrakou et al. (2014) | 2007 | | | | 7.6 | |
| *Top-down estimates* | | | | | | |
| Fu et al. (2007) | 2000 | 4.27 [g] | | 12.7 | | 5.1 |
| Liu et al. (2012) [b] | 2007 | 34.2 | 13.4 | | | |
| Stavrakou et al. (2014) | 2007 | | | | 8.6 | |
| Stavrakou et al. (2015) [c] | 2010 | 20.6 to 24.6 | | | 5.9 to 6.5 | 2.0 to 2.7 |
| Stavrakou et al. (2017) [c] | 2005 | 24.4 | | | 5.8 | |

| | | | | | | |
|---|---|---|---|---|---|---|
| | 2006 | 24.0 | | | (average | |
| | 2007 | 26.7 | | | of | |
| | 2008 | 25.9 | | | emissions | |
| | 2009 | 26.5 | | | from 2005 | |
| | 2010 | 26.1 | | | to 2011) | |
| | 2011 | 25.5 | | | | |
| | 2012 | 25.6 | | | | |
| | 2013 | 27.7 | | | | |
| | 2014 | 27.8 | | | | |
| This work | 2007 | 20.2 [f] (16.4 - 23.6) | 6.5 [f] (5.5 - 7.9) | 19.2 [f] (12.2 - 22.8) | 9.6 [f] (5.4 - 11.7) | 2.48 [f] (2.08 – 3.13) |

[a] These emission estimates included some NMVOC species which were not precursors to formaldehyde or glyoxal and therefore not included in this work. See color keys in Figure 2 for NMVOC species whose emissions were included in this work.

[b] Used SCIAMACHY-observed glyoxal VCDs as constraints.

[c] Used GOME-2A-observed and OMI-observed formaldehyde VCDs as constraints.

[d] Consisted of emissions from open burning of crop residues and from biofuel burning.

[e] Calculated by the GEOS-Chem model using GEOS-5 meteorological data.

[f] Average of top-down estimates from four inversion experiments.

[g] Only anthropogenic emissions of reactive alkenes, formaldehyde, and xylenes from northeastern, northern, central and southern China were included

R2.14    There are many language errors in the manuscript. This decreases its readability. I strongly recommend that the manuscript is corrected by a native speaker among the co-authors and thoroughly re-read.

Thank you for the suggestion. We have carefully rewritten most of the manuscript. The revised manuscript have been proof read by one of the native-English-speaking coauthors.

R2.15    The discussion in Sections 3, 4 is not quantitative. The reader does not get enough information about absolute differences. This should be improved in the revised version.

Thank you for the suggestion. We added Tables S4 to S8 to summarize the statistics of the comparison between satellite-observed and model-simulated formaldehyde and glyoxal VCDs over eastern China. We also added quantitative comparisons in the main text:

[Main text, lines 451 to 459]: The *a priori* simulated formaldehyde VCDs generally reproduced the observed seasonal contrast and spatial patterns over eastern China, with correlation coefficients (R) between 0.74 and 0.94 year-round, except in December (R = 0.51). The *a priori* simulated formaldehyde VCDs were significantly higher than the GOME-2A observations over eastern China between late fall and winter (November, December, January, and February), with normalized mean biases (NMB) of 13% to 67%, implying an overestimate of the anthropogenic formaldehyde precursors in the *a priori* emission estimates. The *a priori* simulated formaldehyde VCDs were lower than the GOME-2A observations over eastern China during May to July (NMB between -11% to -6.4%), implying an underestimation of the emissions of formaldehyde precursors in the *a priori* during May to July.

[Main text, lines 493 to 495]: The *a priori* simulated glyoxal VCDs were generally lower than the GOME-2A glyoxal VCDs over eastern China year-round, especially during the warmer months (NMB between -52% and -59% during May to September, Table S6).

[Main text, lines 519 to 521]: The *a priori* simulated formaldehyde VCDs (at OMI overpass time) were higher than the OMI observations over eastern China year-round (NMB between 22% and 70%, Table S7), suggesting an overestimation of NMVOC emissions year-round.

[Main text, lines 536 to 538]: The *a priori* simulated glyoxal VCDs were lower than the OMI observations throughout the year (NMB between -32% to -66%, Table S8) and especially from March to October, indicating an underestimation of NMVOC sources in the *a priori* year-round.

[Main text, lines 565 to 568]: The optimization was especially effective in optimizing the spatial pattern of the a posteriori formaldehyde VCDs, such that the a posteriori R against the GOME-2A formaldehyde VCDs exceeded 0.85 over eastern China for all twelve months (Table S4).

R2.16    l. 466 : Can you specify what are the differences between the retrievals algorithms? I wonder why you didn't use retrievals from GOME-2 and OMI based on the same retrieval algorithm. These products are available. This should remove undesirable biases due to the different retrieval methodologies.

The Review is correct in pointing out that there are MAX-DOAS retrievals of both GOME-2A and OMI formaldehyde and glyoxal VCDs (De Smedt et al., 2012, 2015; Lerot et al., 2010). However, (1) the GOME-2A and OMI formaldehyde and glyoxal products retrieved using different algorithms provided disparate information on seasonal NMVOC emissions (as shown in Section 3), (2) none of these products have been sufficiently validated over China, and (3) several studies have used these different satellite product to derive top-down emission estimates (e.g., Chan Miller et al., 2016; Stavrakou et al., 2015, 2016). Therefore, the uncertainty associated with the use of different satellite retrievals in top-down Chinese NMVOC emission estimates should be explored in a consistent way.

We emphasized this point in the main text:

[Main text, lines 150 to 153]: In this study, we used satellite retrievals of both formaldehyde and glyoxal, along with a chemical transport model and its adjoint, to constrain NMVOC emissions from China for the year 2007. We conducted sensitivity experiments to evaluate the impacts on the top-down estimates due to different satellite observations, with the goal of bracketing a probable range of top-down estimates.

[Main text, lines 550 to 553]: The qualitative analyses in Section 3 showed that the GOME-2A and OMI retrievals of formaldehyde and glyoxal VCDs provided disparate information on seasonal Chinese NMVOC emissions. Therefore, our four inversion experiments using different satellite observations as constraints represented the range of probable top-down estimates given current satellite observations.

We also provided additional details on the GOME-2A and OMI observations of formaldehyde and glyoxal in Table S2.

[Supplementary information, Table S2]: Technical details for the GOME-2A and OMI formaldehyde and glyoxal observations used in this study

| Technical details | GOME-2A | | OMI | |
| --- | --- | --- | --- | --- |
| | Formaldehyde | Glyoxal | Formaldehyde | Glyoxal |
| Product reference | De Smedt et al. (2012) | Lerot et al. (2010) | González Abad et al. (2015) | Chan Miller et al. (2014) |
| Platform | European MetOp-A satellite | | NASA Aura satellite | |
| Operation time | October 2006 – present | | July 2004 – present | |
| Overpass time | 9:30 local time | | 13:30 local time | |
| Global coverage | Every 1.5 days before June 2013; every 3 days after June 2013 | | Every 1 day | |
| Spatial resolution | 80 km $\times$ 40 km | | 13 km $\times$ 24 km | |
| Spectral window | 240-790 nm | | 270-500 nm | |
| Spectral resolution | 0.26-0.5 nm | | 0.42 nm and 0.63 nm | |

| Selected absorption band | 328.5 - 346 nm | 435 - 460 nm | 328.5 - 356.5 nm | 435 - 461 nm |
|---|---|---|---|---|
| Retrieval algorithm | Differential Optical Absorption Spectroscopy (DOAS) fitting | | Direct fitting | |
| Cloud parameter data | FRESCO+ (Wang et al., 2008) | | OMCLDO2 (Acarreta et al., 2004) | |
| Surface albedo data | Kleipool et al. (2008) | | Kleipool et al. (2008) | |
| Air mass factor calculation | Radiative transfer model | LIDORT (Spurr, 2008) | | VLIDORT (Spurr, 2006) | |
| | Tracer gas profiles | IMAGES model outputs (Stavrakou et al., 2009b) | | GEOS-Chem model outputs (González Abad et al., 2015) | |
| Extinction by aerosols | Considered implicitly via cloud correction (Boersma et al., 2004) | | Considered implicitly in the cloud retrieval (Acarreta et al., 2004) | |
| Discarded pixels | Pixels with cloud fraction >40% or zenith angles >60° were discarded | | Pixels with cloud fraction > 40% were discarded | Pixels flagged as impacted by random telegraph signals were discarded [a] |

[a] Some pixels were flagged as impacted by random telegraph signals in the level 1-B product (Kleipool, 2005).

R2.17    All figures are based on model/data comparisons only for January, April, June, October. By doing that, we miss important information for other months, especially for July and August (maximum of biogenic emissions). The figures are also hard to read. More synthetic figures should be added, for instance showing the monthly variation of the satellite/model columns over large regions.

Thank you for the suggestion. All comparisons (Figures 3 to 10, Tables S4 to S8) between satellite observations, model simulations, and ground-based MAX-DOAS measurements are now presented on a monthly basis, with the exception of measurements at Wuxi, which were only available as bi-monthly means.

We have also increased the figure resolutions and enlarged the symbols in the Figures.

R2.18    Detailed comparisons with ground-based measurements are missing. The ground-based measurements shown in Figures 4-7 leave a lot to be desired. No concrete conclusion can be drawn from these plots with regards to the observed monthly variation and how well the model can reproduce it.

Thank you for pointing out the issue. In response, we added Table S3 to show the details of the MAX-DOAS measurements. We also added the monthly MAX-DOAS measurements of formaldehyde VCDs at Xianghe (a site in the NCP) at GOME-2A and OMI overpass

time, as well as comparisons against satellite observations and model simulations (Figures S4 and S5). We added discussions in the main text.

[Supplementary information, Table S3]: Ground-based MAX-DOAS measurements of formaldehyde and glyoxal vertical column densities in China at GOME-2A and OMI overpass times

| Reference | Location | Time of measurement | | Vertical column densities | |
|---|---|---|---|---|---|
| | | | | 9-10 local time | 13-14 local time |
| Formaldehyde [$10^{16}$ molecules cm$^{-2}$] | | | | | |
| Vlemmix et al. (2015) | Xianghe, Heibei (39.75N, 116.96E) | 2011 | JAN | 0.24 | 0.54 |
| | | | FEB | 0.78 | 0.99 |
| | | | MAR | 0.77 | 0.95 |
| | | | APR | 0.99 | 0.98 |
| | | | MAY | 1.08 | 1.53 |
| | | | JUN | 2.06 | 2.67 |
| | | | JUL | 1.49 | 2.10 |
| | | | AUG | 1.47 | 2.03 |
| | | | SEP | 1.05 | 1.36 |
| | | | OCT | 1.11 | 1.64 |
| | | | NOV | 0.85 | 1.18 |
| | | 2010 | DEC | 0.49 | 0.79 |
| Lee et al. (2015) | Beijing (39.59°N, 116.18°E) | August 16 to September 11, 2006 | | - | 1.79 |
| Wang et al. (2017) | Wuxi, Jiangsu (31.57°N,120.31°E) | 2011 - 2014 | JF | 0.7 [a] | 0.8 [a] |
| | | | MA | 0.9±0.15 [a] | 1.1±0.26 [a] |
| | | | MJ | 1.5±0.12 [a] | 1.9±0.15 [a] |
| | | | JA | 1.7±0.10 [a] | 2.2±0.26 [a] |
| | | | SO | 1.2±0.12 [a] | 1.7±0.12 [a] |
| | | | ND | 0.8±0.30 [a] | 1.4±0.32 [a] |
| Li et al. (2013) | Back Garden, Guangdong (23.50°N, 113.03°E) | July 2006 | | 1.3±1.0 [b] | 1.3±0.7 [b] |
| Glyoxal [$10^{14}$ molecules cm$^{-2}$] | | | | | |
| Li et al. (2013) | Back Garden, Guangdong | July 2006 | | 6.8±5.2 [c] | 11.4±6.8 [c] |

| | (23.50ºN, 113.03ºE) | | | |
|---|---|---|---|---|

[a] From Figure 12 of Wang et al. (2017)

[b] From Figure 4 of Li et al. (2013)

[c] From Figure 5 of Li et al. (2013)

[Supplementary information, Figure S4]:

[Figure]

**Figure S4. Measured and simulated monthly mean formaldehyde VCDs at Xianghe at GOME-2A overpass time: MAX-DOAS measurements (black line, Vlemmix et al., 2015), GOME-2A measurements (green solid line), GOME-2A measurements multiplied by 1.7 (blue solid line), monthly mean formaldehyde VCDs from the *a priori* simulation (red line), the IE-1 *a posteriori* simulation (green dashed line), and the IE-3 *a posteriori* simulation (blue dashed line). Pearson correlation coefficients (R) of the satellite-observed and simulated formaldehyde VCDs against the MAX-DOAS measurements are shown in the top left. Annual mean bias (MB, in units of $10^{15}$ molecules cm$^{-2}$) of the satellite-observed and simulated formaldehyde VCDs against the MAX-DOAS measurements are shown in the bottom right.**

[Supplementary information, Figure S5]:

[Figure]

**Figure S5 Measured and simulated monthly mean formaldehyde VCDs at Xianghe at OMI overpass time: MAX-DOAS measurements (black line, Vlemmix et al., 2015), OMI measurements (green solid line), monthly mean formaldehyde VCDs from the *a priori* simulation (red line), the IE-2 *a posteriori* simulation (green dashed line), and the IE-4 *a posteriori* simulation (blue dashed line). Pearson correlation coefficients (R) of the satellite-observed and simulated formaldehyde VCDs against the MAX-DOAS measurements are shown in the top left. Annual mean bias (MB, in units of $10^{15}$ molecules $cm^{-2}$) of the satellite-observed and simulated formaldehyde VCDs against the MAX-DOAS measurements are shown in the bottom right.**

[revised manuscript text omitted]

---

## Author Comment (AC2) · 30 Jun 2018

Please see the supplement for our reply for both reviews on acp-2017-1136 (Hansen Cao et al., 2018)

Please also note the supplement to this comment:
https://www.atmos-chem-phys-discuss.net/acp-2017-1136/acp-2017-1136-AC2-supplement.pdf

---

## Author Response (AR2)

We thank the second reviewer for the constructive and detailed comments. In response, we have carefully edited the text to improve clarity throughout. We respond to each specific comment below. The reviewer's original comments are shown in red. Our replies are shown in black. The corresponding changes in the manuscript are shown in blue.

Reviewer 2:

The revised manuscript by Cao et al. represents a major improvement compared to the previous version. The authors have delivered a serious effort to comply with the criticism of the first report by including comparisons with ground-based observations, and by significantly improving the presentation and consistency. However, there are still weaknesses in the revised manuscript. Page numbers refer to the revised version.

1) Section 3 should present only comparisons with the a priori model, but a posteriori results are also shown in Figure S4 in the same section. This is inconsistent with the section title, please change for consistency.

Thanks for pointing this out. We understand that the inclusion of the *a posteriori* results in Figures S4 and S5 may cause some confusion in Section 3. However, it was necessary to also show the *a posteriori* model simulation in Figures S4 and S5 to facilitate the visual comparisons between observations, the *a priori*, and the *a posteriori* results in Section 4 (lines 638-648). If we split the a posteriori results into separate figures, there would be a lot of repetition between the figures, and it would be much harder for the readers to make the visual comparison. To reduce the potential confusion to readers, we revised the main text to improve clarity:

Main text: line 475 to line 490: Figure S4 compares the GOME-2A and the model a priori formaldehyde VCDs in 2007 against the multi-year (during the years 2010 to 2016) monthly mean formaldehyde VCD measured by MAX-DOAS at Xianghe (a rural site in the NCP) at GOME-2A overpass time (Vlemmix et al., 2015). The GOME-2A formaldehyde VCDs were consistent with the MAX-DOAS measurements in terms of the seasonal variation (R = 0.95) but showed an annual mean bias of -3.78 $\times 10^{15}$ molecules cm$^{-2}$. The interannual variability of the local formaldehyde VCDs (as represented by the standard deviation of the MAX-DOAS measurements) was relatively small and thus unlikely to be sole driver for the differences between the GOME-2A observations in 2007 and the MAX-DOAS measurements during 2010 to 2016. The seasonal variation of the model a priori formaldehyde VCDs were less consistent with that of the MAX-DOAS measurements (R = 0.81). Figure S4 also showed that, by multiplying the GOME-2A formaldehyde VCD observations by 1.7, the annual mean bias against the MAX-DOAS measurements at Xianghe was reduced to -0.21 $\times 10^{15}$ molecules cm$^{-2}$. Figures 3 and 4 show that the differences between the satellite and MAX-DOAS measurements were also reduced at Wuxi when the GOME-2A formaldehyde VCDs were scaled up by 1.7. These findings offered some support for using the GOME-2A formaldehyde VCDs scaled by 1.7 as an upper-bound constraint for Chinese NMVOC emissions.

Main text: line 638 to line 648: Figure S4 also compared the model a posteriori formaldehyde VCDs in 2007 against the GOME-2 observations, the model a priori formaldehyde VCDs, and the MAX-DOAS measurements (during 2010-2016) at Xianghe at GOME-2 crossing time. Compared to the a priori, our a posteriori formaldehyde VCDs were in better agreement with the seasonal variation of the MAX-DOAS measurements (R values

increased from 0.81 for the a priori to 0.95 for IE-1 and 0.93 for IE-3). During the warm months (May to September), the monthly a posteriori formaldehyde VCDs from IE-1 and IE-3 bracketed the interannual variation of monthly formaldehyde VCDs measured by MAX-DOAS. For the rest of the year, both the GOME-2A observations and the a posteriori formaldehyde VCDs were systematically biased low relative to the MAX-DOAS measurements. As discussed before, these biases could not be fully accounted by the interannual variability of the local formaldehyde VCDs and was thus likely due to sampling or retrieval difference between the MAX-DOAS and the satellite.

Supplementary information: Figure S4

[Figure]

Figure S4. Measured and simulated monthly mean formaldehyde VCDs at Xianghe at GOME-2A overpass time: MAX-DOAS measurements (black line, monthly mean averages for the years 2010 to 2016 from Vlemmix et al., 2015), GOME-2A measurements (green solid line), GOME-2A measurements multiplied by 1.7 (blue solid line), monthly mean formaldehyde VCDs from the a priori simulation (red line), the IE-1 a posteriori simulation (green dashed line), and the IE-3 a posteriori simulation (blue dashed line). Pearson correlation coefficients (R) of the satellite-observed and simulated formaldehyde VCDs against the MAX-DOAS measurements are shown in the top left. Annual mean biases (MB, in units of $10^{15}$ molecules $cm^{-2}$) of the satellite-observed and simulated formaldehyde VCDs against the MAX-DOAS measurements are shown in the bottom right.

Supplementary information: Figure S5

[Figure]

Figure S5 Measured and simulated monthly mean formaldehyde VCDs at Xianghe at OMI overpass time: MAX-DOAS measurements (black line, monthly mean averages for the years 2010 to 2016 from Vlemmix et al., 2015), OMI measurements (green solid line), monthly mean formaldehyde VCDs from the a priori simulation (red line), the IE-2 a posteriori simulation (green dashed line), and the IE-4 a posteriori simulation (blue dashed line). Pearson correlation coefficients (R) of the satellite-observed and simulated formaldehyde VCDs against the MAX-DOAS measurements are shown in the top left. Annual mean biases (MB, in units of $10^{15}$ molecules $cm^{-2}$) of the satellite-observed and simulated formaldehyde VCDs against the MAX-DOAS measurements are shown in the bottom right.

2) The ground-based data are not obtained during the studied year (2007). How can this affect the comparisons? The interannual variability of CH2O MAX-DOAS data should be discussed. For stations with ground-based data available for more than one year, the model results should be compared to a mean seasonal cycle.

Thank you for the suggestion. As suggested, we compared the satellite-observed and the simulated formaldehyde VCDs against the multi-year (2010 to 2016) monthly average and standard deviations of the MAX-DOAS formaldehyde measurements at Xianghe to represent the multi-year mean seasonal cycle and the interannual variability of the local formaldehyde VCDs. As shown in Table S3, Figure S4, and Figure S5, the differences between the satellite and the MAX-DOAS observations could not be fully accounted by the interannual variability of the local formaldehyde VCDs.

Main text: line 475 to line 490: Figure S4 compares the GOME-2A and the model a priori formaldehyde VCDs in 2007 against the multi-year (during the years 2010 to 2016) monthly

mean formaldehyde VCD measured by MAX-DOAS at Xianghe (a rural site in the NCP) at GOME-2A overpass time (Vlemmix et al., 2015). The GOME-2A formaldehyde VCDs were consistent with the MAX-DOAS measurements in terms of the seasonal variation (R = 0.95) but showed an annual mean bias of -3.78 $\times 10^{15}$ molecules cm$^{-2}$. The interannual variability of the local formaldehyde VCDs (as represented by the standard deviation of the MAX-DOAS measurements) was relatively small and thus unlikely to be sole driver for the differences between the GOME-2A observations in 2007 and the MAX-DOAS measurements during 2010 to 2016. The seasonal variation of the model a priori formaldehyde VCDs were less consistent with that of the MAX-DOAS measurements (R = 0.81). Figure S4 also showed that, by multiplying the GOME-2A formaldehyde VCD observations by 1.7, the annual mean bias against the MAX-DOAS measurements at Xianghe was reduced to -0.21 $\times 10^{15}$ molecules cm$^{-2}$. Figures 3 and 4 show that the differences between the satellite and MAX-DOAS measurements were also reduced at Wuxi when the GOME-2A formaldehyde VCDs were scaled up by 1.7. These findings offered some support for using the GOME-2A formaldehyde VCDs scaled by 1.7 as an upper-bound constraint for Chinese NMVOC emissions.

Main text: line 638 to line 648: Figure S4 also compared the model a posteriori formaldehyde VCDs in 2007 against the GOME-2 observations, the model a priori formaldehyde VCDs, and the MAX-DOAS measurements (during 2010-2016) at Xianghe at GOME-2 crossing time. Compared to the a priori, our a posteriori formaldehyde VCDs were in better agreement with the seasonal variation of the MAX-DOAS measurements (R values increased from 0.81 for the a priori to 0.95 for IE-1 and 0.93 for IE-3). During the warm months (May to September), the monthly a posteriori formaldehyde VCDs from IE-1 and IE-3 bracketed the interannual variation of monthly formaldehyde VCDs measured by MAX-DOAS. For the rest of the year, both the GOME-2A observations and the a posteriori formaldehyde VCDs were systematically biased low relative to the MAX-DOAS measurements. As discussed before, these biases could not be fully accounted by the interannual variability of the local formaldehyde VCDs and was thus likely due to sampling or retrieval difference between the MAX-DOAS and the satellite.

Supplementary information: Table S3 Ground-based MAX-DOAS measurements of formaldehyde and glyoxal vertical column densities in China at GOME-2A and OMI overpass times

| Reference | Location | Time of measurement | Vertical column densities | |
|---|---|---|---|---|
| | | | 9-10 local time | 13-14 local time |
| Formaldehyde [$10^{16}$ molecules cm$^{-2}$] | | | | |
| Vlemmix et al. (2015) | Xianghe, Hebei (39.75N, 116.96E) | 2010-2016 | JAN 0.51±0.17 | 0.82±0.17 |
| | | | FEB 0.70±0.07 | 0.95±0.04 |
| | | | MAR 0.89±0.12 | 1.12±0.17 |
| | | | APR 1.04±0.11 | 1.21±0.16 |
| | | | MAY 1.39±0.19 | 1.87±0.28 |
| | | | JUN 1.86±0.25 | 2.41±0.24 |
| | | | JUL 1.75±0.27 | 2.16±0.15 |
| | | | AUG 1.67±0.20 | 2.18±0.22 |

| | | | | | |
|---|---|---|---|---|---|
| | | | SEP | 1.24 ±0.21 | 1.57 ±0.17 |
| | | | OCT | 1.17 ±0.15 | 1.49 ±0.14 |
| | | | NOV | 0.80 ±0.03 | 1.11 ±0.15 |
| | | | DEC | 0.61 ±0.13 | 0.89 ±0.07 |
| Lee et al. (2015) | Beijing (39.59°N, 116.18°E) | August 16 to September 11, 2006 | | - | 1.79 |
| Wang et al. (2017) | Wuxi, Jiangsu (31.57°N,120.31°E) | 2011 – 2014 | JF | 0.7 [a] | 0.8 [a] |
| | | | MA | 0.9 ±0.15 [a] | 1.1 ±0.26 [a] |
| | | | MJ | 1.5 ±0.12 [a] | 1.9 ±0.15 [a] |
| | | | JA | 1.7 ±0.10 [a] | 2.2 ±0.26 [a] |
| | | | SO | 1.2 ±0.12 [a] | 1.7 ±0.12 [a] |
| | | | ND | 0.8 ±0.30 [a] | 1.4 ±0.32 [a] |
| Li et al. (2013) | Back Garden, Guangdong (23.50°N, 113.03°E) | July 2006 | | 1.3 ±1.0 [b] | 1.3 ±0.7 [b] |
| Glyoxal [$10^{14}$ molecules cm$^{-2}$] | | | | | |
| Li et al. (2013) | Back Garden, Guangdong (23.50°N, 113.03°E) | July 2006 | | 6.8 ±5.2 [c] | 11.4 ±6.8 [c] |

[a] From Figure 12 of Wang et al. (2017)
[b] From Figure 4 of Li et al. (2013)
[c] From Figure 5 of Li et al. (2013)

3) l.467. 'Nevertheless, these ground-based measurements....', change to 'The measurements...' to avoid repetition with l. 465. Here and elsewhere check the text to avoid repetitions.

Thanks for pointing out this repetition. We changed the expression to:

Main text: line 465 to line 472: In principle, these ground-based measurements are not directly comparable to the satellite-observed and model-simulated formaldehyde VCDs, due to the coarse spatial resolution of our analyses. Nevertheless, the MAX-DOAS measurements showed that (1) formaldehyde VCDs were higher during the warmer months relative to the colder months; (2) formaldehyde VCDs over Wuxi (in central eastern China) were higher than those over Xianghe (in northern China) and Back Garden (in southern China) for most months; (3) in June, the formaldehyde VCDs over Xianghe were the highest among the three MAX-DOAS sites, reflecting the strong emissions from biomass burning in the NCP.

4) l.446. 'reflecting the seasonal biomass burning emissions there', change to 'reflecting the occurrence of seasonal biomass burning'

Changed as suggested. Thank you.

Main text: line 445 to line 446: In spring, GOME-2A formaldehyde VCDs were high over Southwest China and Southeast Asia, reflecting the occurrence of seasonal biomass burning.

5) l.542-548. Needs rewriting. Why 'two possible causes'? Change to 'Possible causes for this apparent contradiction could be...' What kind of errors do you mean in l.545? In the chemical oxidation scheme? Please elaborate on the possible errors.

Thanks for pointing out this lack of clarity. We re-wrote the sentences.

Main text: line 550 to line 556: It thus appeared that the constraints on Chinese NMVOC emissions indicated by the OMI formaldehyde and glyoxal observations were contradictory. Possible causes for this apparent contradiction could be: (1) the chemical production and losses of formaldehyde and glyoxal at different times of the day were not accurately simulated by the model, which would also explain why the MAX-DOAS measurements of formaldehyde and glyoxal VCDs were both higher in the afternoon than in the morning, while the model showed an opposite diurnal contrast; and (2) it is also possible that there were different inherent biases in the OMI formaldehyde and glyoxal retrievals.

6) l.553. 'disparate, and apparently contradictory'

Changed the the main text as suggested, thankyou.

Main text: line 560 to line 562: The qualitative analyses in Section 3 showed that the GOME-2A and OMI retrievals of formaldehyde and glyoxal VCDs provided disparate, and apparently contradictory information on seasonal Chinese NMVOC emissions.

7) l.559. You mention that the cost function decrease is 8% in one inversion. This reduction is unusually modest. Can you explain in which inversion experiments this occurred and what were the reasons?

Thanks. The one case where the cost function was reduced by only 8% occurred in the optimization in IE-3 for April. In that case, the initial cost function was already small. We added the explaination in the main text.

Main text: line 567 to line 571: Relative to their respective initial cost function values, the optimized cost function values were reduced by 8% to 75% for all four experiments. The unusually modest 8% reduction occurred in the optimization in IE-3 for April. In that case, the initial cost function value was small; i.e., the a priori formaldehyde VCDs were already in good agreement with 1.7 times the GOME-2A formaldehyde VCDs (Figure 3 and Table S5).

8) l.617. 'The impacts...was', correct the verb. This type of error exists in other sentences too, please check carefully the grammar before resubmission.

Thanks for pointing out this type of error. We corrected it thoughout the whole main text.

Main text: line 629 to line 630: The impacts of satellite glyoxal observations on constraining Chinese glyoxal precursors emission estimates were further demonstrated in IE-4.

9) l.742. 'seasonal contrast', replace by 'seasonal amplitude'

Thank you for the suggestion. We revised the wording here to improve clarity and to comply with this comment and the next comment.

Main text: line 764 to line 769: As discussed above, three out of our four inversion experiments showed a stronger summer-versus-winter contrast in the NMVOC emissions, compared to the a priori emissions (Figure 2). We evaluated the impacts of this stronger seasonal amplitude in NMVOC emissions on surface ozone and secondary organic carbon (SOC) aerosol concentrations by driving the GEOS-Chem model with the a priori NMVOC emission estimates and with the average top-down emission estimates from our four inversion experiments, respectively.

10) l.744. 'stronger seasonal contrast' appears also in l.742. Try to avoid repetitions throughout the text.

Thank you for point this out. We revised the wording here to improve clarity and to avoid repetition.

Main text: line 764 to line 769: As discussed above, three out of our four inversion experiments showed a stronger summer-versus-winter contrast in the NMVOC emissions, compared to the a priori emissions (Figure 2). We evaluated the impacts of this stronger seasonal amplitude in NMVOC emissions on surface ozone and secondary organic carbon (SOC) aerosol concentrations by driving the GEOS-Chem model with the a priori NMVOC emission estimates and with the average top-down emission estimates from our four inversion experiments, respectively.

11) l.763-765. Bad phrasing. Start your sentence by e.g. 'The comparisons for ozone corroborate the stronger seasonal amplitude of the top-down NMVOC emissions derived in this study'.

Thanks. We re-wrote the sentence.

Main text: line 788 to line 789: These comparisons for surface ozone corroborated the stronger seasonal amplitude of the top-down NMVOC emissions derived in this study.

12) Section 6. Here comparisons are presented for June and December. Why did you focus only on those months? The year-round ozone variability at sites where data are available should be checked.

We focused on June and December because these were the two months when the differences between our averaged top-down emission estimate and the a priori emission estimate were the greatest. We wanted to evaluate the impacts of top-down VOCs emission on the surface ozone production during these two extreme conditions. Another difficulty was that regionally-representative surface ozone measurements were surprisingly scarce in China for all the other months in the literature. We explain our reasoning in the text:

Main text: line 773 to line 775: We focused here on surface ozone in June and December, when the differences in NMVOC emissions between our averaged top-down estimate and the a priori emission estimate were greatest.

13) Table 1. The last row seems to be equal to the average of the previous rows. Why don't you write this explicitly in the last row instead of 'Our top-down estimates'?

Changed as suggested, thank you.

*Table 1 Inversion experiments to constrain Chinese NMVOC emissions*

| Inversion experiments | Observational constraints from satellites [±uncertainties] | Annual Chinese NMVOC emission estimates [Tg y⁻¹] | | | |
|---|---|---|---|---|---|
| | | Anthropogenic | Biogenic | Biomass burning | Total |
| | | *A priori* emission estimates [uncertainty] | | | |
| | | 18.8 (5.4 for aromatics) [a] [factor of two uncertainty] | 17.3 (7.5 for isoprene) [b] [±55% uncertaitny | 2.27 [factor of three uncertainty] [c] | 38.3 |
| | | *A posteriori* emission estimates [range of estimates] | | | |
| IE-1 | GOME-2A formaldehyde [±90%] and glyoxal [±150%] | 17.8 (5.8 for aromatics) | 20.0 (9.8 for isoprene) | 2.27 | 40.1 |
| IE-2 | OMI formaldehyde [±90%] and glyoxal [±150%] | 16.4 (5.5 for aromatics) | 12.2 (5.4 for isoprene) | 2.08 | 30.7 |
| IE-3 | GOME-2A formaldehyde ×170% [±90%] | 23.6 (6.6 for aromatics) | 22.8 (11.3 for isoprene) | 3.13 | 49.5 |
| IE-4 | OMI glyoxal [±150%] | 23.0 (7.9 for aromatics) | 21.6 (11.7 for isoprene) | 2.43 | 47.0 |
| Average top-down estimates | | 20.2 (6.5 for aromatics) | 19.2 (9.6 for isoprene) | 2.48 | 41.9 |

[a] From Li et al. (2017)

[b] From Guenther et al. (2006).

[c] Compiled from the emission estimated by van der Werf et al. (2010) plus a scaling of the emission estimated by Huang et al. (2012). See text (section 2.2) for details.